# Translation factor eIF5a is essential for IFNγ production and cell cycle regulation in primary CD8⁺ T lymphocytes

Thomas C. J. Tan [1], Van Kelly [2], Xiaoyan Zou[1], David Wright [1], Tony Ly [2,3] & Rose Zamoyska [1]✉

Control of mRNA translation adjusts protein production rapidly and facilitates local cellular responses to environmental conditions. Traditionally initiation of translation is considered to be a major translational control point, however, control of peptide elongation is also important. Here we show that the function of the elongation factor, eIF5a, is regulated dynamically in naïve CD8⁺ T cells upon activation by post-translational modification, whereupon it facilitates translation of specific subsets of proteins. eIF5a is essential for long-term survival of effector CD8⁺ T cells and sequencing of nascent polypeptides indicates that the production of proteins which regulate proliferation and key effector functions, particularly the production of IFNγ and less acutely TNF production and cytotoxicity, is dependent on the presence of functional eIF5a. Control of translation in multiple immune cell lineages is required to co-ordinate immune responses and these data illustrate that translational elongation contributes to post-transcriptional regulons important for the control of inflammation.

Regulation of protein expression at the level of ribosome production and translation can rapidly rewire and co-ordinate responses to changing environmental cues. Poor correlations between transcriptomes and proteomes have been documented in numerous cell types, including stem cells[1] and T lymphocytes[2] indicating that a substantial proportion of their protein production is regulated translationally. Both these lineages can undergo rapid expansions and changes in cell identity in response to environmental triggers[3]. Regulation of ribosome biogenesis and global mRNA translation[3] contribute to this plasticity together with more targeted post-transcriptional regulons, important for directing the production of proteins with related functions or involved in common biological pathways[4]. Regulation of such subsets of mRNAs frequently involves interactions with RNA-binding proteins or non-coding RNAs (eg miRNA and lncRNA)[5,6]. In addition, alterations in ribosome constituents or the proteins involved with the translation of mRNAs can

influence protein production. We showed that for naïve CD8⁺ T cells the degree of ribosome biogenesis was dramatically affected by the extent of activation through the T cell receptor (TCR), which impacted translation both globally and selectively for groups of proteins[7]. How this plasticity of translational control is orchestrated around the ribosome is not well understood, but a number of proteins are directly involved with the ribosome during translation[8,9] and these may themselves be regulated both translationally and post-translationally. One such protein is the translation factor eIF5a, one of the top 20 most abundant proteins in CD8⁺ T cells[2]. eIF5a was originally identified as a translation initiation factor, but may also function in translation elongation and termination[10] for specific subsets of proteins[11]. In particular, eIF5a is required for efficient peptide bond formation between consecutive proline residues and may influence translation of glycine and charged amino acids[10,12], suggesting it is an important player in influencing post-transcriptional regulons.

[1]Institute of Immunology and Infection Research, University of Edinburgh, Ashworth Laboratories, Charlotte Auerbach Road, Edinburgh EH9 3FL, UK.
[2]Wellcome Trust Centre for Cell Biology, University of Edinburgh, Michael Swann Building, Max Born Crescent, Edinburgh EH9 3BF, UK. [3]Centre for Gene Regulation and Expression, Life Sciences Research Complex, University of Dundee, Dundee DD1 5EH, UK. ✉e-mail: Rose.Zamoyska@ed.ac.uk

eIF5a isoforms in eukaryotes and archaea are the only proteins with post-translational attachment of a hypusine residue on Lys[50-51] [13], a modification that is essential for eIF5a function[14]. The site of hypusination is in contact with the acceptor stem of the P site tRNA on 80S ribosome[15] and non-modified eIF5a was inefficient in facilitating peptide synthesis[16], while inhibition of the hypusination pathway impaired proliferation in mammalian cells[17,18]. Spermidine, a natural polyamine, is the donor moiety for eIF5a hypusination and is reduced in ageing cells. Dietary supplementation of spermidine restored both the abundance of eIF5a hypusination and age-related impairments in T cell function[19], and extended the life span of yeast, flies, worms, and cultured human PBMCs[20]. Therefore strong evidence links the abundance of eIF5a hypusination and its function in age-related immunosenescence.

The aim of this study was to determine first how the activity of eIF5a itself is regulated in T lymphocytes, and secondly, whether eIF5a selectively regulates translation of specific subsets of mRNA during T cell activation. We use CRISPR knockout of eIF5a itself and of the dedicated enzymes required for its hypusination, deoxyhypusine synthetase (DHPS) and deoxyhypusine hydroxylase (DOHH), and compared the effect of the knockouts with inhibition of hypusination by the deoxyhypusine synthase inhibitor, GC7. We show that naïve T cells express abundant eIF5a but with limited hypusine modification suggesting restricted functionality. Upon activation eIF5a protein production is up-regulated and DHPS and DOHH enzymes are expressed making the hypusinated form of eIF5a more abundant. Total loss of eIF5a surprisingly increased production of some ribosomal proteins, translation elongation factor eEF1 components and a possible translational inhibitor, eIF6, while decreasing components of translation initiation factors eIF3 and eIF5, suggesting changes in the composition of the mRNA translation machinery. Mass spectrometry analysis of nascent peptide production in eIF5a knockout T cells confirmed unequal impacts on synthesis of specific sets of proteins in the absence of eIF5a, particularly those involved in cell cycle regulation and cytokine production, with the most profound impact on IFNγ production. These acute changes were accompanied by a failure of the cells to survive in the longer term. Treatment with GC7 did not faithfully reproduce the phenotypes of DHPS and DOHH knockout cells indicating a significant off-target impact of this drug. Overall our data are consistent with eIF5a co-ordinating post-transcriptional regulons in T lymphocytes and thus primary T cell function.

## Results

### T cell activation promotes maturation of eIF5a
A unique feature of T cells is that they exist for long periods as naïve cells which are functionally immature, non-dividing and metabolically inert, with very low levels of macromolecular synthesis[21]. Upon stimulation through their antigen-specific TCRs they rapidly turn on gene transcription and protein synthesis. In the initial 24 h following stimulation CD8[+] T cells increase cytoplasmic mass by approximately 4-fold, indicating a rapid initiation of macromolecular synthesis, before commencing a burst of proliferation with a division time as short as 2–4h[22,23].

We examined the functional maturation of eIF5a in OT-1 CD8[+] T cells which express a single specificity transgenic TCR that recognises a peptide, SIINFEKL, from chicken ovalbumin. Western blot of cell lysates from naïve OT-1 T cells showed abundant expression of eIF5a, but low signal with a hypusine-specific Ab (Fig. 1a, b). Upon activation the abundance of hypusinated eIF5a increased by ~3-fold, concurrent with a similar increase in expression of DHPS, the enzyme that catalyses transfer of the 4-aminobutyl moiety of spermidine to eIF5a. The abundance of both hypusinated eIF5a and DHPS peaked between 24-48 h, increasing ~2-fold above that of a control protein, ZAP-70. Single-cell measurement of hypusinated and total eIF5a using flow cytometry (Fig. 1c, d) corroborated the western blot and hypusine

and eIF5a abundance was shown to increase by 4 h. Staining peaked at 24 h with ~7-fold increase in hypusination and 4-fold increase in total eIF5a, a 1.5-2 fold greater increase of hypusine relative to total eIF5a, which made it likely that both newly synthesised and pre-existing pools of eIF5a were hypusinated. [3]H-spermidine was shown previously to label a single protein with a hypusine modification upon activation of human lymphocytes[13]. The protein was subsequently identified as eIF5a (previously eIF-4D[24]) and it is the sole protein in the eukaryote proteome to carry the hypusine modification.

### Abrogation of eIF5a disrupts key cellular functions
Hypusination of eIF5a allows efficient facilitation of translation elongation in vitro[16] and may be responsible for its cytoplasmic localisation[25]. However, it is unclear whether or not the non-hypusinated precursor and deoxyhypusinated intermediate forms of eIF5a have any activity in primary cells. This question is important as targeting DHPS or DOHH, as a means to inhibit eIF5a, may not be an optimal strategy given that deletion of budding yeast DOHH had only a mild phenotype[26] whereas embryonic targeting of DOHH in mouse was lethal[27]. Many studies utilise the drug GC7 to inhibit hypusination and function of eIF5a. GC7 is a spermidine derivative which competes with spermidine for binding to DHPS, preventing the conversion of non-hypusinated eIF5a precursor to its deoxyhypusinated intermediate, but it is undocumented whether GC7 has effects beyond inhibiting eIF5a.

Naïve OT-1 T cells were activated for 2 days, and then either treated with GC7 for 2 days or transfected with CRISPR-Cas9 guides to disrupt *Eif5a*, *Dhps* or *Dohh* genes. *Eif5a* KO cells were routinely recovered after 3d culture in cytokine IL-2 to maximise cell recovery while other KOs could be left for 4d. Time-course analysis further showed that the phenotypic changes of the knockouts, measured by flow cytometry with antibodies against specific proteins (representative gating strategies shown in Supplementary Fig. 1), correlated with loss of the CRISPR targeted molecules (Supplementary Fig. 2a). CRIPSR-Cas9 transfection controls were targeted for the surface molecule, Thy 1. Given targeting efficiency was maximally 80%, both KO and untargeted cells from the same culture could be analysed by intracellular staining for eIF5a or hypusine to identify positive cells. Figure 2a shows a representative experiment with a mean percentage KO of *Eif5a*, *Dhps*, and *Dohh* of 38.4%, 71.6% and 78.9%, respectively. Depletion of eIF5a protein in *Eif5a* KO cells was confirmed by western blotting using FACS-sorted eIF5a positive and negative populations (Fig. 2b). eIF5a depletion has been shown to affect cellular proliferation[11,28], and this was reflected by the significantly lower cell recovery from cultures in *Eif5a* KO cells compared to control *Thy1* KO cells. Similarly there was a decrease in cell recovery when hypusination was impaired in GC7-treated, *Dhps* and *Dohh* KO cells but this was not as profound as for the *Eif5a* KO (Fig. 2a).

Cellular proliferation was examined in greater detail using Hoechst staining to identify phases of the cell cycle. Surprisingly GC7 inhibition of eIF5a hypusination resulted in cell cycle profiles that were distinct from those observed following knockout of *Eif5a*, *Dhps*, or *Dohh*. GC7 caused a pronounced cellular accumulation in S phase whereas *Eif5a* and *Dhps* KO cells tended to accumulate in the G0/1 phase of the cell cycle while knockout of *Dohh* showed no appreciable changes in the cell cycle profile (Fig. 2a) despite reduced hypusination. Thus loss of eIF5a and DHPS likely caused prolonged G1 or defective G0-1 entry, while the deoxyhypusinated intermediate form of eIF5a in DOHH KOs, was partially active possibly slowing progression through the cell cycle. In contrast GC7-treated cells accumulated in S phase suggesting that the drug has off-target effects on cell cycle beyond its impact on eIF5a hypusination.

Spermidine is a key substrate for hypusination and function of eIF5a, and its loss impacts the autophagic flux in both B and T cells. Compared to young individuals, T and B cells from elderly people have

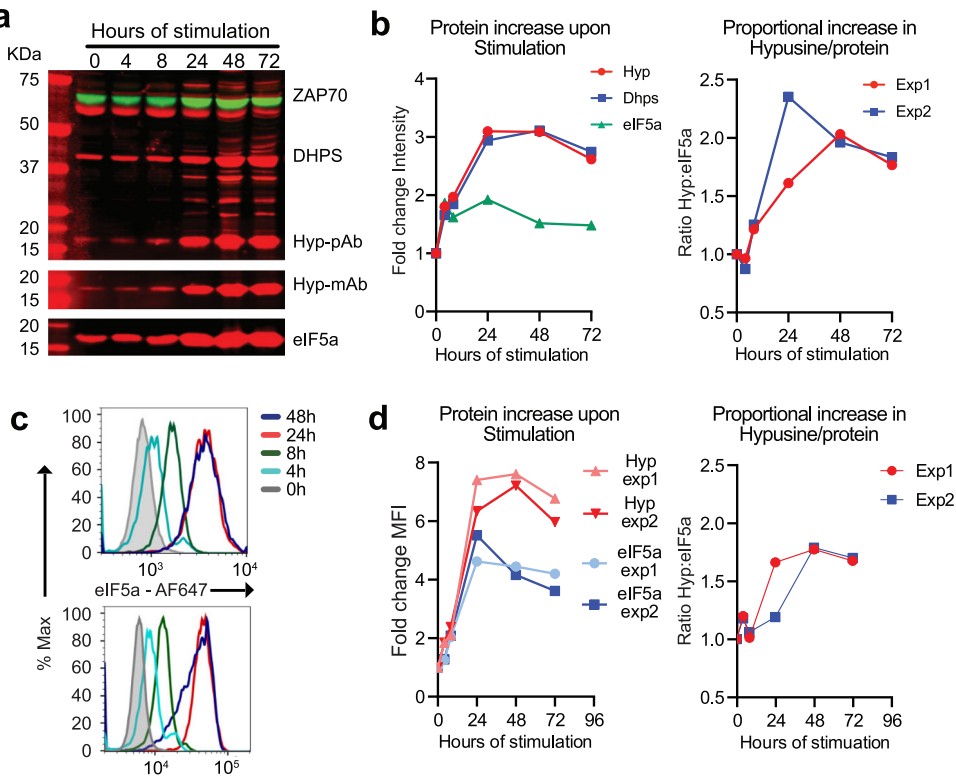

**Fig. 1 | Hypusination of eIF5a is induced by T cell activation. a** Naïve OT-1 lymph node CD8[+] T cells were stimulated with agonist peptide SIINFEKL for the times indicated and a western blot of total cell lysates was probed with Abs for DHPS, hypusine, eIF5a and ZAP70 as loading control, equal cell numbers were loaded in each lane. **b** Fold up-regulation and Hyp:eIF5a ratio were calculated relative to the 0 h time point, after normalisation against ZAP70 intensity at the same time point. One representative of three independent experiments is shown. Lines in left panel: red, Hyp-pAb; blue, DHPS; green: eIF5a. Lines in right panel: red, Experiment 1; blue, Experiment 2. **c** Flow cytometric analysis over time are shown of OT-1 cells stained

with Abs against eIF5a and hypusine. The small brighter peak of staining at 4 h may be an artefact due to aggregation of cells which could not be excluded by single cell gating strategies. Lines: grey shaded, 0 h; light blue, 4 h; green, 8 h; red, 24 h; dark blue, 48 h. **d** Relative mean fluorescent intensities (MFI) from two independent flow cytometry experiments are shown and were used to calculate the Hyp:eIF5a ratio on the right. Lines in left panel: light and dark red, Hypusine in Experiment 1 and 2, respectively; light and dark blue, eIF5a in Experiment 1 and 2, respectively. Lines in right panel: red, Experiment 1; blue, Experiment 2.

reduced autophagy which was reversed by the provision of spermidine through a mechanism involving eIF5a supported translation of the master transcription factor, TFEB, which regulates autophagy[29,30]. KO CD8 CTL were treated with the lysosomal inhibitor bafilomycin A1 (BafA1) or vehicle for 2 h and selectively permeabilised to allow detection of autophagosomal membrane-bound LC3-II[30]. Autophagic flux was defined as the MFI of LC3-II in BafA1-treated cells divided by vehicle-treated cells. Whereas GC7-treatment and *Eif5a* KO reduced autophagy to a similar extent, KO of *Dhps* and *Dohh* did not affect autophagic flux (Fig. 2c). These data confirm that eIF5a is important for autophagy but appear to contradict data which showed that autophagy was reduced in *Dohh* KO cells[30]. The latter measured autophagy in immortalised KO MEFs so the discrepancy may reflect residual eIF5a hypusination sufficient to maintain the autophagic flux in our experiments or may lie with the different cell lineages used for the analyses.

After 3-4d culture in IL-2 T cells differentiate to cytotoxic T lymphocytes (CTL) and we tested the impact of loss of either hypusination or eIF5a protein on effector function. KO cells were restimulated with antigenic peptide for 4 h to reactivate effector cytokine transcription and translation and labelled with puromycin during the last 20 min to assess global protein production. Incorporation of puromycin into nascent proteins was analysed by flow cytometry and showed significant reduction following all treatments (Fig. 3a, b). In contrast, the production of a specific effector cytokine IFNγ, encoded by *Ifng* gene, was most profoundly affected in *Eif5a* KO cells, with the majority of the

population falling in the negative gate, and was less reduced in GC7 treated and *Dhps* KO cells while *Dohh* KO cells showed only a minor decrease in IFNγ[+] cells. Control *Thy1* KO cells showed no reduction in either total protein or specific cytokine production. Cytokine production was also followed over time and showed loss of IFNγ protein mirrored loss of eIF5a (Supplementary Fig. 2a). In contrast, TNF production by *Eif5a* KO cells was comparable to control cells at d2 and showed only a small reduction by d3. Together these data indicate that the loss of IFNγ production was selective rather than reflecting a general shut down in cytokine translation.

We assessed whether transcription or translation of *Ifng* and *Tnf* mRNA was impacted by the KOs, by performing RT-qPCR analysis on cell lysates from GC7/mock-treated cells and unseparated *Thy1*, *Eif5a*, *Dhps* and *Dohh* targeted populations. In these experiments two separate CRISPR guide RNAs were used to target *Eif5a* increasing the KO efficiency to >80%, comparable to *Dhps* and *Dohh* targeted cultures. In contrast to the non-restimulated control cells, all cells restimulated with peptide significantly up-regulated *Ifng* and *Tnf* mRNA as expected (Fig. 3c). *Dhps*, or *Dohh* KO or GC7-treated cells did not show decreased *Ifng* and *Tnf* mRNAs compared to control cells while KO of eIF5a reduced *Ifng* mRNA by approximately 30% and surprisingly increasing production of *Tnf* mRNA. We conclude that fully matured eIF5a is required for translation of *Ifng* and to a lesser extent *Tnf*, but that loss of eIF5a also impacts transcription for example, of *Ifng*, likely via reduced translation of eIF5a-dependent transcription factors essential for cytokine production, as discussed below.

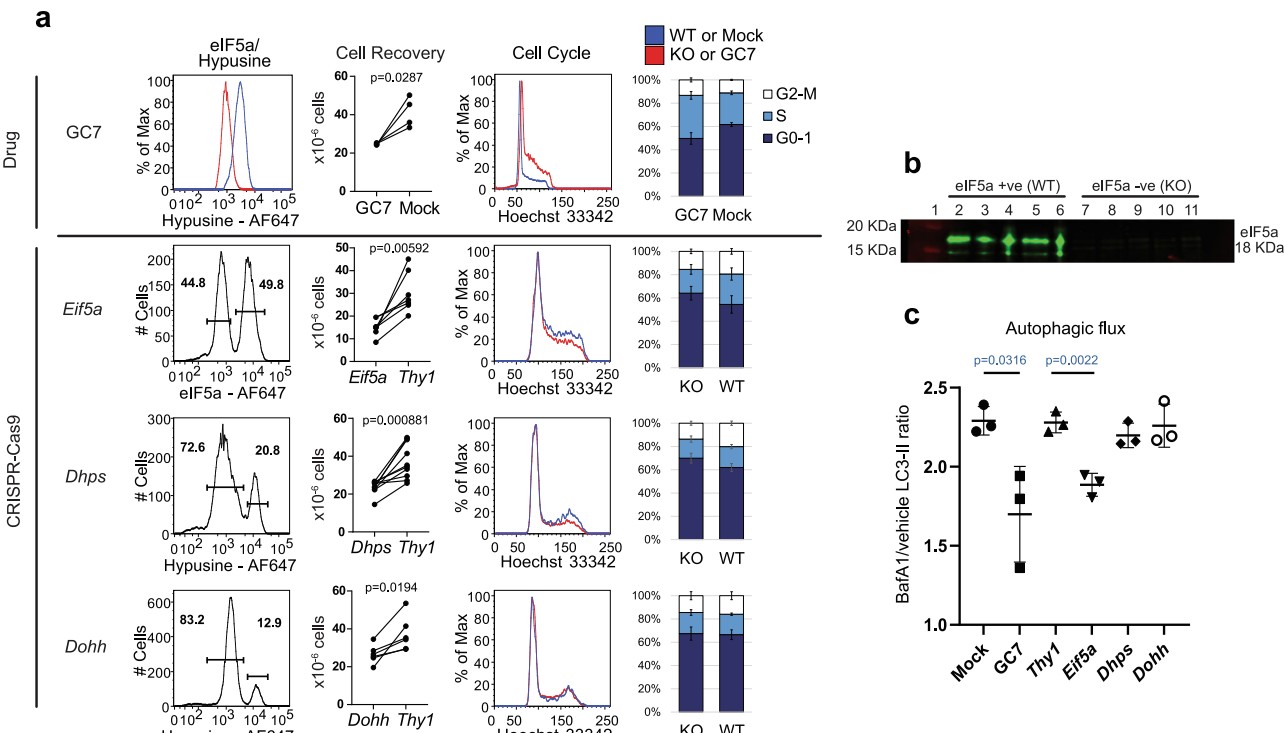

**Fig. 2 | eIF5a regulates cell cycle and autophagy. a** FACS overlay histograms showing expression of eIF5a or hypusine assessed at d3 (GC7 and *Eif5a*) and d4 (*Dhps* and *Dohh*). KO of cell surface marker Thy1 was included as a CRISPR control. Cell Recovery column shows the number of cells recovered at the time of harvest for each treatment paired to their control treatment (mock-treatment for GC7 and *Thy1* KO for CRISPR cells, *n* = 4 independent experiments for GC7, 7 for *Eif5a*, 10 for *Dhps*, 6 for *Dohh*). Cell cycle profiles were obtained from Hoechst 33342 staining. Mean percentage of cell cycle phases was plotted, pairing positive and negative cells from individual cultures (*n* = 4 independent experiments for GC7, 12 for *Eif5a*, 7 for *Dhps*, 8 for *Dohh*). Error bars represent SD between experiments. P values were calculated using a paired two-sided *t* test. Lines: blue, WT or Mock; red, KO or GC7. Bars: dark blue, G0-1; light blue, S; white, G2-M. **b** Western blot of *Eif5a*-CRISPR cells

sorted into eIF5a positive and negative populations by flow cytometry according to their eIF5a intensities, and detected with an eIF5a antibody (*n* = 5 biological replicates). Lane 1: size marker; lane 2-6: sorted eIF5a+ cells (WT); lane 7-11: sorted eIF5a-cells (KO). **c** Autophagic flux was quantified by measuring the level of autophagosomal membrane-bound LC3-II in cells treated with lysosomal inhibitor bafilomycin A1 (BafA1) compared to mock vehicle-treated cells (*n* = 3 biological samples, *P* values were calculated using unpaired two-sided *t* test) Centre line and error bars represent mean value ±SD between biological replicates. **a–c** OT-1 LN T cells were stimulated for 48 h with 100 nM SIINFELK peptide before CRISPR-Cas9 KO of *Eif5a*, *Dhps* or *Dohh* or treatment with 10 μM GC7 and were subsequently maintained in IL-2. The GC7 experiment with its controls was performed independently of the CRISPR-KO experiment.

We tested whether *Eif5a*, *Dhps* and/or *Dohh* KO CTL killed antigen-expressing target cells (Fig. 3d). There was a three-fold reduction in specific killing after knockout of *Eif5a* while loss of *Dohh* had no impact and the *Dhps* knockout showed a slight improvement in killing in two independent experiments. *Thy1* KO cells also consistently killed targets more effectively which was likely due to Thy1 being a highly glycosylated surface molecule which may negatively impact effector:target cell conjugation. The reduction in the killing capacity of *Eif5a* KO cells was not due to changes in the production of the key cytotoxic effector molecule, Granzyme B, as its production was unaffected by loss of eIF5a, nor did it appear to be due to exhaustion of the CTL as PD-1 expression was unchanged (Supplementary Fig. 2a).

We further asked whether KO cells could respond to infection following challenges in vivo. OT-1 *Eif5a*, *Dhps* or *Dohh* targeted cells (expressing CD45.1) were each mixed 1:1 with *Thy1* KO cells (expressing CD45.2) and injected into wild type recipients together with ovalbumin expressing *Listeria monocytogenes* (*Lm*OVA). Splenocytes were harvested 6d later and the frequency of recovered donor cells assessed (Supplementary Fig. 2b). Very few targeted cells were recovered from the infected mice compared to the Thy1-KO control cells, irrespective of whether eIF5a was knocked out directly or its hypusination was impaired. Those CD45.1 knockout cells that were recovered were exclusively cells that had failed to delete the gene of interest (Supplementary Fig. 2b) indicating that for CD8 CTL cell expansion and/or survival in vivo the expression of eIF5a and its hypusination are essential.

## Systematic analysis of eIF5a-regulated translation in T cells

The regulation of specific groups of proteins by functional eIF5a was assessed by incorporation of 4-Azido-L-homoalanine (AHA) in place of methionine into newly synthesised proteins which globally labels the nascent peptide pool. The advantage of this approach is that it directly measures the synthesis of new proteins in order to characterise the functions of translation factors more sensitively than total proteomics, however, it may not be as comprehensive at measuring mRNA translation as conventional techniques such as ribosome profiling. *Eif5a* KO and GC7-treated cultures were pulsed with AHA for 2 h and *Eif5a* KO and WT populations were separated by cell sorting for eIF5a following fixation and staining, ensuring that untargeted WT cells were subjected to identical culture and labelling conditions as the KO cells. Following lysis of the sorted cells, AHA-labelled peptides were isolated using copper-mediated Click chemistry with Alkyne agarose beads and nascent proteomes were identified and quantified using liquid chromatography-mass spectrometry, LC-MS.

Single shot label-free proteomics of 12 fractions of pooled GC7- and mock-treated cells resulted in detection of 7529 proteins which served as a spectral library for all experimental mass-spec samples. Using this library, 5953 proteins were identified as represented by more than 2 unique peptides (compared to 3682 proteins detected without the library). 3246 proteins were expressed in at least 3 out of 4 biological samples in all conditions (Supplementary Data 1). All these proteins were significantly enriched (≥5x more abundant) in AHA-

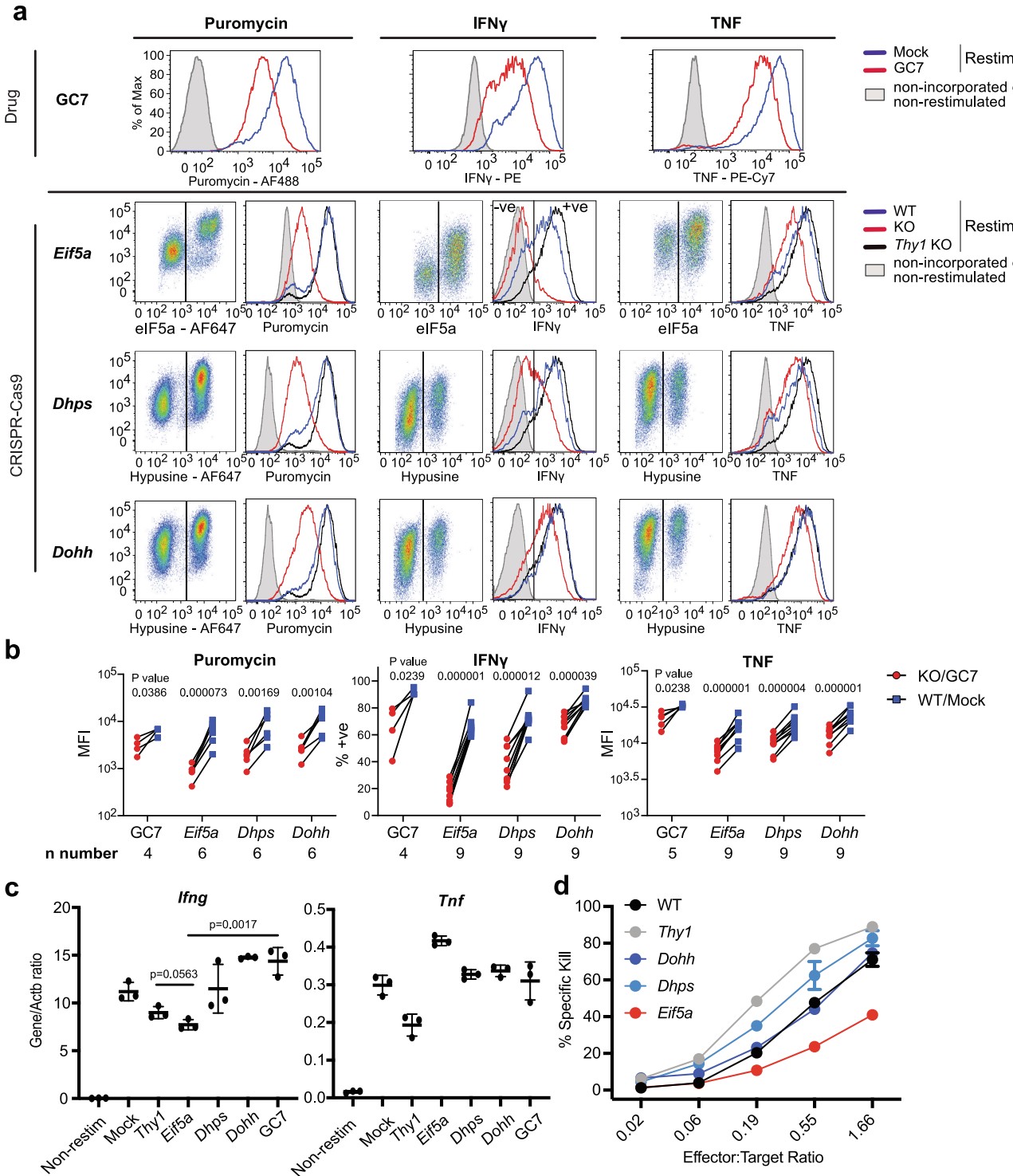

**Fig. 3 | eIF5a regulates IFNγ translation and cytotoxicity. a** FACS dot plots indicate gating for positive and negative cells and histograms show puromycin incorporation, IFNγ and TNF expression as indicated in GC7 or CRISPR KO cells which were restimulated with 100 nM SIINFEKL peptide and Brefeldin A for 4 h prior to staining. Lines: grey shaded, non-incorporated or non-restimulated control; blue, WT or Mock; red, KO or GC7. **b** Mean fluorescent intensity (MFI) of paired samples gated for positive and negative cells as shown in (**a**) from >4 independent experiments (individual *n* numbers stated on graph. *P* values were calculated using paired two-sided *t* test). Dots: blue, WT or Mock; red, KO or GC7. **c** *Ifng* and *Tnf*

mRNA was measured by RT-qPCR from bulk drug-treated or transfected cells as indicated (*n* = 3 biological samples, *P* value were calculated using unpaired two-sided *t* test). Centre line and error bars represent mean value ±SD between biological replicates. **d** Specific cytotoxic activity of T cell KO and control populations was quantified by FACS by changes in the ratio between EL-4 cells pulsed with SIINFEKL peptide and control unpulsed, fluorescently labelled EL-4 cells (2 independent experiments with 3 technical replicates each). Error bars represent SD between technical replicates. Lines and dots: black, WT; grey, *Thy1*; dark blue, *Dohh*; light blue, *Dhps*; red, *Eif5a*.

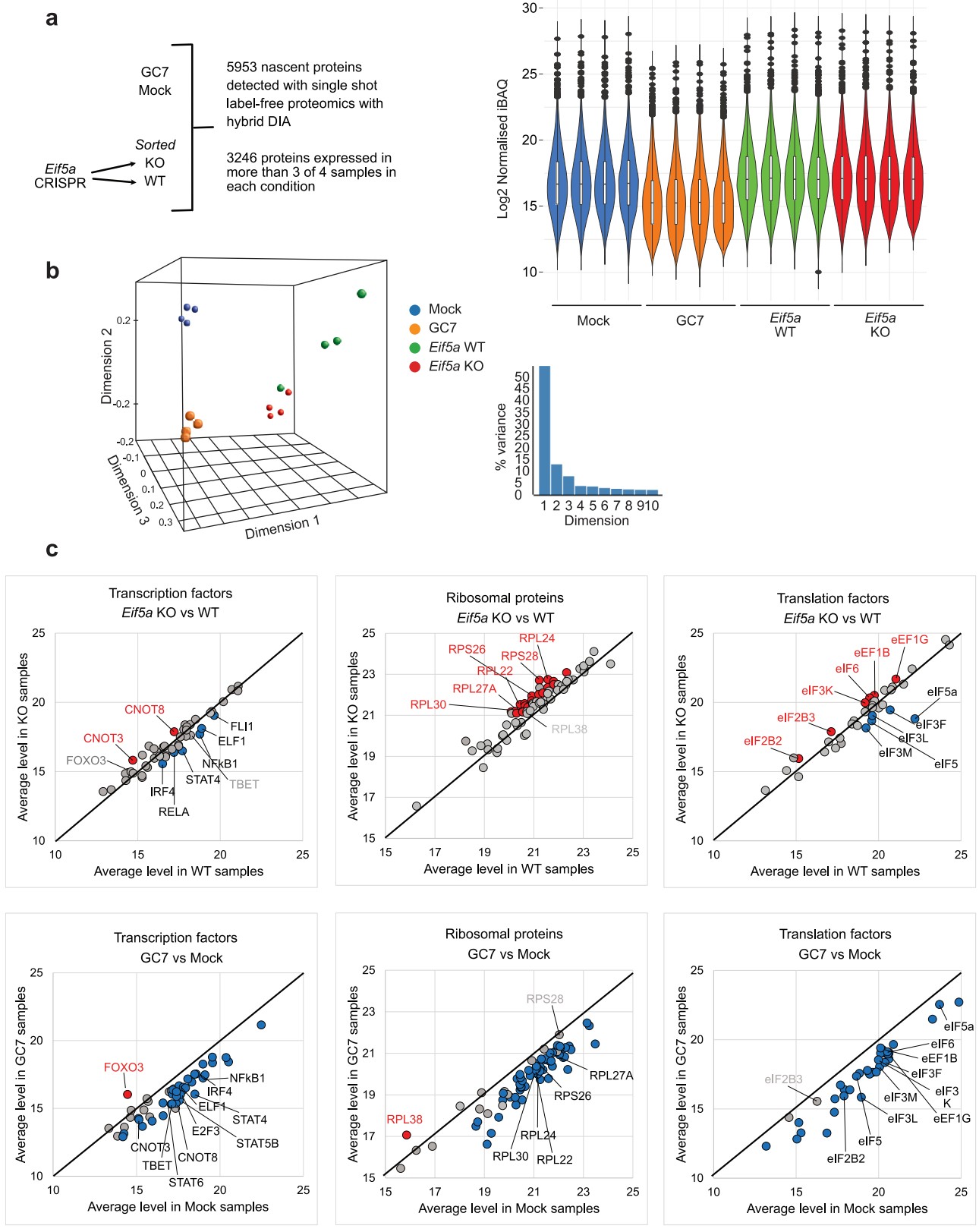

labelled samples compared to the non-labelled background. Four biological replicate samples were used for each group (Fig. 4a, left). iBAQ values were used to measure absolute quantities of protein and to compare between groups[31]. Blank in one sample was replaced by the average value of the other 3 samples. The iBAQ values were normalised by Vsn algorithm[32] on NormalyzerDE[33] and log2-transformed, followed

by a second normalisation according to the input cell number to allow comparison between groups (Fig. 4a, right; Supplementary Fig. 3a, b; Supplementary Data 1). Amongst the four sample groups (Mock, GC7, *Eif5a* KO and WT), nascent proteins in GC7-treated cells were globally reduced by ~3-fold. This decrease in AHA incorporation was consistent with previous trial proteomic experiments using various ratios

**Fig. 4 | Systematic analysis of GC7- or *Eif5a* KO-affected nascent proteomes.**
**a** Schematic indicating the groups analysed and the number of proteins identified. In total 5953 proteins (left) were detected, of which 3246 were detected in more than 3 of 4 samples in all groups (*n* = 4 biological replicates). (Right) iBAQ values from each group were normalised using Vsn algorithm, and then by the input cell number. Log2-transformed values of 3246 proteins in each sample after normalisation were used to represent protein abundance and plotted as violin plots. Center line, median; box limits, upper and lower quartiles; whiskers, 1.5x interquartile range; points, outliers. Colours: blue, Mock; orange, GC7; green, *Eif5a* WT; red, *Eif5a* KO. **b** 3D-MDS plot was used to visualise the difference between samples (left). The percentage of variance represented by each dimension is plotted on the right. **c** Group average of normalised protein levels for all detected transcription factors, ribosomal proteins, and translation factors were compared between KO

and WT cells or between GC7- and mock-treated cells. Significantly (for KO-WT, paired two-sided *t* test *P* value <0.05, absolute FC > 1.5; for GC7-Mock, unpaired two-sided *t* test with Benjamini–Hochberg correction FDR < 0.05, absolute FC > 1.5) changed proteins of interest are annotated in black (down-regulated) or red (up-regulated) font while proteins not deemed significant by the statistical test, but may still be biologically relevant are annotated in grey font. OT-1 T cells were stimulated with peptide for 48 h before CRISPR-Cas9 KO of *Eif5a* or incubation with GC7. Cells were labelled with AHA for the final 2 h of culture. *Eif5a* KO cells and untargeted (WT) controls were obtained from the same culture using FACS sorting. AHA-labelled proteins were isolated by Click-chemistry with alkyne agarose beads and digested with trypsin. Detailed data handling procedure is described in Supplementary Fig. 3a (*n* = 4 biological samples).

between GC7- and mock-treated input cells (Supplementary Fig. 3c), and therefore is unlikely to be caused by the input normalisation. 3D MDS plot of all nascent proteins detected between *Eif5a* KO and WT cells and between GC7-treated and mock-treated control cells (Fig. 4b) showed distinctive signatures separating the 4 groups with the largest source of variations (Dimension 1) separating KO/WT from GC7/Mock, most likely due to formaldehyde fixation and the subsequent reverse crosslinking in the KO/WT samples. The second largest source of variation (Dimension 2) separated KO from WT and GC7 from Mock and is most likely reflecting manipulation of eIF5a activity. Differential expression analyses were conducted separately between GC7- and mock-treated cells and between *Eif5a* KO and WT cells. GC7 had a systematic effect on protein production, with 2617 proteins down-regulated and only 5 proteins up-regulated (Supplementary Fig. 4a); while *Eif5a* KO had a more limited effect on protein production, with 234 down-regulated and 216 up-regulated proteins (Supplementary Fig. 4b).

Clustering of GO terms using WebGestalt[34] compared the down- and up-regulated proteins to all MS-detected newly synthesised proteins as background (Supplementary Fig. 4a, b). Noticeably transcription factors, ribosomal proteins and translation factors were affected by both GC7 and *Eif5a* KO but in different ways (Fig. 4c). Several key regulators of environmental responses, cytokine and effector T cell functions, including IRF4, ELF1, FLI1, NFκB1, STAT1, STAT3, STAT4, STAT5A, STAT5B, STAT6, and TBET (*Tbx21*) were down-regulated in their production in GC7-treated cells, while in *Eif5a* KO cells only IRF4, ELF1, NFκB1, STAT4, and to a lesser extent FLI1 and TBET were negatively impacted. Production of cell cycle regulator E2F3[35] was down in GC7-treated cells but not in *Eif5a* KO cells, despite both populations having impaired proliferation. In contrast, the NFκB transcription factor RELA which is important for survival of proliferating T cells[36] was significantly down-regulated in *Eif5a* KO cells but not in GC7-treated cells. Some transcription factors were up-regulated, such as the memory cell survival factor FOXO3; while some were down-regulated in GC7-treated cells but up-regulated in *Eif5a* KO cells, such as CNOT3 and CNOT8, components of the CCR4-CNOT deadenylase complex which destabilises its target RNAs including cell cycle regulators[37,38].

Given the profound effect of *Eif5a* KO on puromycin incorporation, we investigated components of the cellular translation machinery in the nascent proteome. Amongst the 71 detected ribosomal proteins, 20 were up-regulated in *Eif5a* KO cells (Supplementary Data 1) while none was down-regulated. Up-regulation of specific ribosomal proteins suggest that ribosomes with alternative composition might be produced in cells in the absence of eIF5a which may preferentially translate a different subset of mRNAs compared to those in WT cells[39,40]. Additionally, down-regulation of translation initiation factors eIF3 components and eIF5 suggest the formation and function of the 43 S pre-initiation complex may be reduced[41], while an increase in eIF6 would prevent the 60 S ribosome subunit from joining with the 40 S subunit[42]. Together these data illustrate a general reduction in protein production, and an altered translation profile in *Eif5a* KO cells.

All the aforementioned proteins were down-regulated upon GC7 treatment, illustrating broader inhibition of protein synthesis in these samples. Polyamines are known to modulate both the efficiency and fidelity of protein synthesis by direct interaction with the ribosome[43]. It is possible that the eIF5a-independent effects of GC7 may result from its competitive binding to ribosomes displacing other polyamines present in the cell.

The direct effects of eIF5a on translation versus indirect effects following changes in transcription factor production, were examined by next generation mRNA-seq of GC7- and mock-treated samples (Supplementary Fig. 4c and Supplementary Data 2). Although GC7-treatment has off-target effects independent of eIF5a, these populations provided suitable quality RNA unlike RNA extracted from formaldehyde-fixed, sorted eIF5a KO cells. Translationally down-regulated genes in the GC7-treated samples were defined as the 1417 genes with no significant reduction in RNA abundance but decreased in their nascent proteins (Fig. 5a, left Venn diagram and Supplementary Data 3). These genes were overlaid with those reduced in their nascent proteins in *Eif5a* KO compared to WT cells (Fig. 5a, right Venn diagram, Supplementary Data 3) giving 86 genes whose translation, not transcription, we could confidently assign as being dependent on eIF5a. Proteins translationally down-regulated only by GC7 treatment were enriched in GO terms: structural constituent of ribosome, mRNA binding, rRNA binding, and translation factor activity (Fig. 5b); while the 86 proteins translationally reduced by both GC7 treatment and *Eif5a* KO did not significantly cluster into any GO terms, although some key genes regulating cellular functions such as T cell receptor signalling transducer FYN, cytokine receptor IL2RG, mTORC1 modulators SLC3A2 and SLC7A5, DNA replication factor RFC3/5, transcription factor IRF4, and translation factors eIF3F/M were in this Venn group (Supplementary Data 3).

Given published evidence that eIF5a facilitates translation through polyproline and charged amino acid regions of proteins, the frequencies of previously characterised eIF5a-controlled 3-mer motifs (PPP, PPG, DVG, DDG, GGT, and RDK[10]) were counted in CDS of the eIF5a-regulated gene set (Fig. 5c). Compared to the frequencies of 50 randomly selected 3-mer sequences, PPP is enriched. PPG and DVG were possibly enriched but rejected by the statistic test (*P* > 0.99). To assess the polyproline content of these genes, they were overlaid with a published list of genes in the mouse genome with PPP or PPG motifs in their coding region[44]. We plotted the percentage of genes containing polyproline in 2900 randomly selected genes in the transcriptome (repeated 1000 times), all expressed proteins detected by LC-MS, and translationally down-regulated genes in GC7-treated or *Eif5a* KO cells (Fig. 5d). In GC7-treated cells, the percentage of polyproline genes was similar between all expressed and translationally down-regulated proteins, suggesting no enrichment for polyproline while the translationally down-regulated proteins in *Eif5a* KO cells had a noticeable increase in polyproline content (Fig. 5d), consistent with a more targeted inhibition by eIF5a in the KO versus GC7-treated samples.

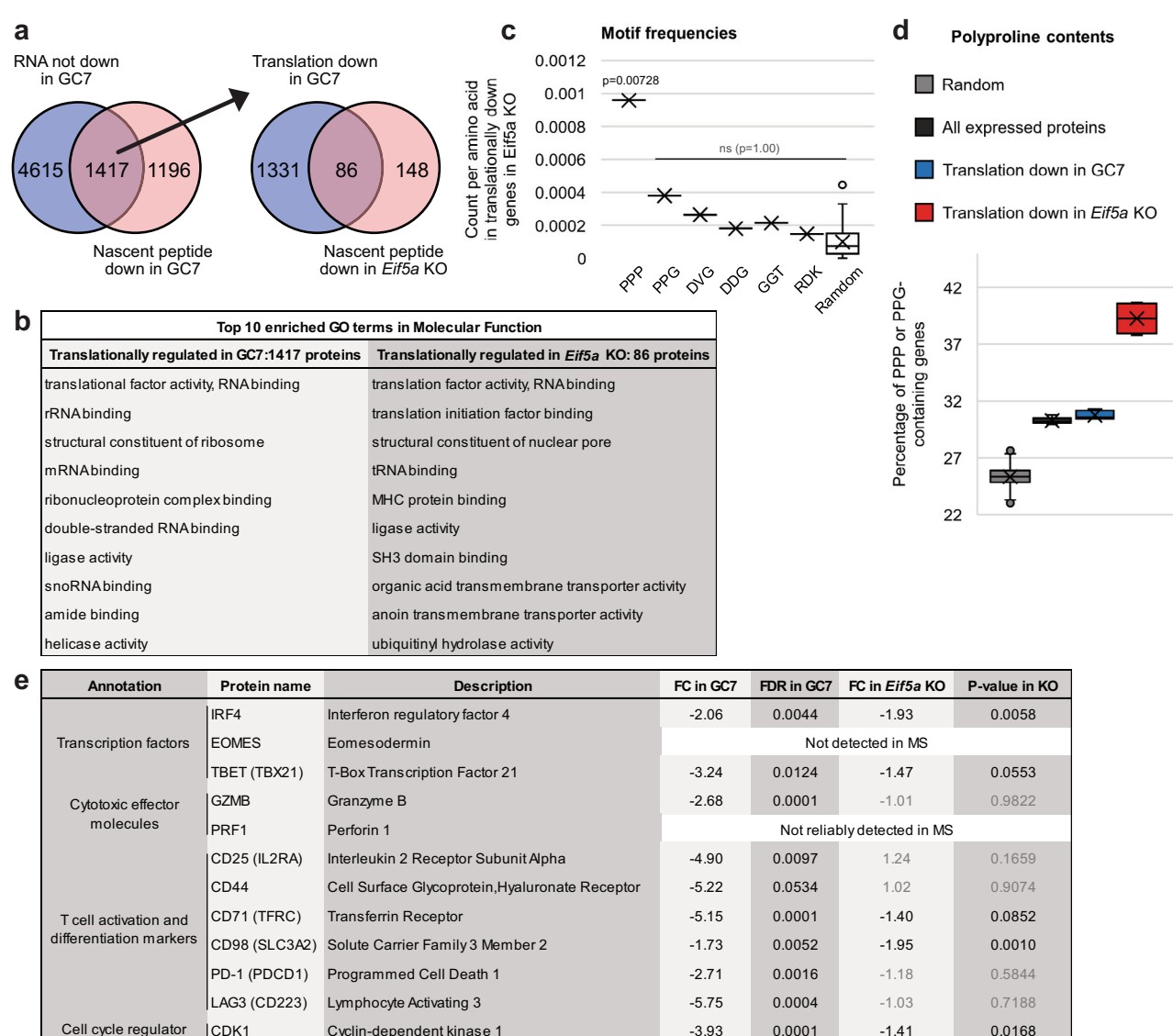

**Fig. 5 | Identification of translationally regulated genes affected by GC7 treatment and *Eif5a* KO. a** Venn diagram overlays identify 1417 proteins not reduced in their mRNA (FDR > 0.1 or FDR < 0.01 and FC > 2, edgeR, *n* = 4, red dots in Supplementary Fig. 4c right) following GC7 treatment but down-regulated in their nascent peptides (FDR < 0.05, FC < −1.5, unpaired two-sided *t* test with Benjamini−Hochberg correction, *n* = 4 biological samples). These proteins were further overlaid with those reduced in their nascent proteins in *Eif5a* KO cells (*P* < 0.05, FC < −1.5, paired two-sided *t* test, *n* = 4 biological samples) as an extra criterion of selection for proteins translationally regulated by eIF5a (86 proteins). Lists of genes in these Venn groups are in Supplementary data 3. **b** Top ten enriched GO terms in molecular function were listed for proteins translationally affected by GC7-treatment or by *Eif5a* KO. **c** Frequencies of specific 3mer sequence motifs in the eIF5a-regulated gene set, normalised by the total length of these proteins. Normalised counts of 50 randomly generated 3mer sequences is plotted at the

right. *P* values were calculated using one-sided Fisher's exact test. **d** Box plots of the percentage of PPP or PPG-containing genes in 2900 randomly selected genes in the transcriptome (repeated 1000 times) (grey), all expressed proteins (black, *n* = 8 biological replicates), translationally down-regulated genes in GC7-treated (blue, *n* = 4 biological replicates) or *Eif5a* KO (red, *n* = 4 biological replicates) cells. **e** Selected eIF5a-regulated proteins involved in regulation of T cell cytotoxic function, differentiation, cell cycle progression and cytokine production from the datasets are listed and their fold changes in nascent proteins in GC7-treated and *Eif5a* KO cells are shown. FDRs were calculated using unpaired two-sided *t* test with Benjamini−Hochberg correction, *P* values were calculated using paired two-sided *t* test. Value colours: black, significant or likely significant; grey, not significant. For all box-plots: cross, mean; center line, median; box limits, upper and lower quartiles; whiskers, 1.5x interquartile range; points, outliers.

We then focused our analyses and validation on some of the proteins involved in regulating T cell differentiation, cytokine production and cell cycle progression whose translation we had identified as affected by the availability of mature eIF5a (listed in Fig. 5e), since defects of these cellular functions were observed in *Eif5a* KO cells. Moreover, these cellular functions are the major age-related impairments described in CD8[+] T cells[45], and suggest impaired function of eIF5a may contribute to immunosenescence. Proteins were measured in mock-treated, GC7-treated, and *Eif5a*-, *Dhps*- and *Dohh*-KO cells using flow cytometry (Supplementary Fig. 2a and 5a).

Key results are summarised from Day3 *Eif5a*-KO and Day4 *Dhps*- and *Dohh*-KO cells (Fig. 6) and across 4 days of culture after treatment (Supplementary Fig. 5a). The abundance of mRNA for these genes were also measured by qRT-PCR (Supplementary Fig. 5b). IRF4, TBET, and EOMES are transcription factors regulating production of IFNγ and TNF in T cells[46–48], and CDK1 interacts with different cyclins to facilitate cell cycle progression[49]. In agreement with the systematic datasets, the abundance of IRF4 and TBET dropped in *Eif5a* KO, *Dhps* KO and to a lesser extent in *Dohh* KO cells, with no reduction in mRNA levels (Fig. 6b) which likely contributed to the decrease in *Ifng*

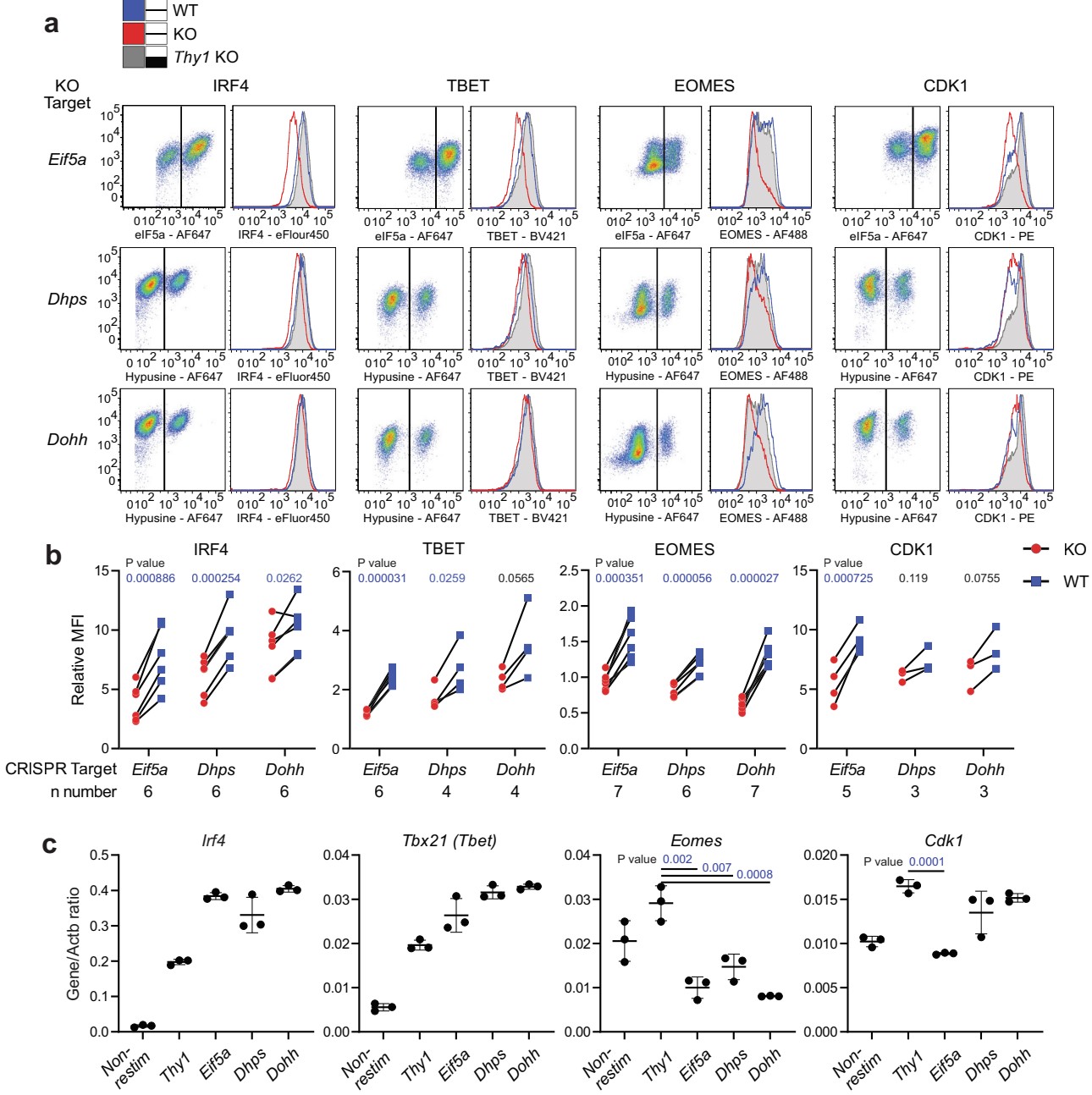

**Fig. 6 | Flow cytometry and qRT-PCR validation of eIF5a-regulated proteins.** **a** Intracellular stains for eIF5a or hypusine were used to identify WT (blue line) and KO (red line) populations in *Eif5a*, *Dhps*, or *Dohh* CRISPR-KO cells (dot plots on left show the extent of KO). *Thy1* CRISPR cells (grey shaded line) were used as a KO control. Overlay histograms on right show IRF4, TBET, EOMES and CDK1 expression in electronically gated WT and KO samples. **b** Quantification of multiple paired samples from the analysis in (**a**) (*P* values were calculated using paired two-sided *t* test). Dots: blue, WT; red, KO. **c** mRNA of these genes were quantified by RT-qPCR, normalised by the level of Actb gene, in fresh bulk transfected cells (*n* = 3 biological samples, *P* values were calculated using unpaired two-sided *t* test). Centre line and error bars represent mean value ±SD between biological replicates, respectively. *P* value colours: blue, significant; black, non-significant.

mRNA in *eIF5a* KO cells. In contrast, all KO cells had decreased levels of EOMES, both in protein and RNA. CDK1 proteins decreased in *Eif5a* KO, but not significantly in *Dhps* KO and *Dohh* KO cells, in correlation with their RNA levels and their cell cycle profiles in Fig. 2a. Curiously cell surface levels of CD98 were increased in GC7-treated cells, which may reflect an impact of GC7 on recycling of this protein, and was apparently an eIF5a-independent effect of the drug since the increase was not observed in any CRISPR cells. Overall, our validated targets agreed with the systematic datasets while also highlighting some difference in RNA dynamics between GC7-treated cells and CRISPR cells.

## Spermidine availability affects eIF5a-regulated proteins

The polyamine spermidine is produced naturally in cells and its abundance decreases with age. Exogenous supplementation of spermidine has been shown to increase organismal longevity[20] and restore age-related T cell functions in an autophagy-dependent manner[19]. In addition to being required for hypusination of eIF5a, spermidine is also known for inhibiting the acetyltransferase activity of Ep300 and thereby inducing autophagy[50]. This non-eIF5a-dependent activity of spermidine may have contributed to discrepancies between cellular phenotypes of GC7 (a spermidine derivative) treated cells and *Eif5a/ Dhps* KO cells. The influence of spermidine on eIF5a function in

activated T cells was examined, by adding an ornithine decarboxylase inhibitor DFMO to the cultures for 48 h to block spermidine biosynthesis. Additionally, DMFO-treated cells were supplemented with exogenous spermidine to overcome the block or were treated with GC7.

Quantitation of eIF5a hypusine levels reflected the predicted outcome of these treatments. The DFMO-treated cells had decreased hypusine, which was restored to a level comparable to untreated cells by spermidine supplementation. Depletion of hypusination by DFMO was exacerbated by additional GC7 treatment (Fig. 7a). However, when we examined the abundance of total and selected proteins whose production we identified as being sensitive to loss of eIF5a by flow cytometry, an altered pattern emerged (Fig. 7b). Puromycin incorporation and CDK1, TBET, IRF4, and IFNγ abundance decreased following DFMO treatment, and were restored by exogenous spermidine addition, indicating they are very sensitive to cellular spermidine levels. Combining DMSO with GC7 further decreased Puromycin incorporation, and expression of CDK1 and IFNγ but did not further reduce the abundance of TBET and IRF4 protein. Interestingly, although IFNγ production was affected by DFMO + GC7 and DFMO, as reported recently for CD8⁺ T cells[29], the protein was still detectable in all cells, as opposed to *EifSa* and *Dhps* KO cells where a significant proportion of the population produced no detectable IFNγ (Fig. 2b). In contrast, TNF, CD25 and CDC45 showed no decrease in expression following DMSO treatment but were reduced by combining DFMO with GC7, indicating that they are less sensitive to spermidine depletion (Fig. 7) despite showing some regulation by eIF5a (Supplementary Data 1).

Together these data indicate that the maintenance of some cellular proteins such as CDK1, TBET and IRF4 are very sensitive to the availability of functional hypusinated eIF5a. On the other hand, other proteins such as the cytokine IFNγ, requires total ablation of eIF5a or DHPS in order to show a profound reduction in abundance.

## Discussion

Our current understanding of eIF5a function largely comes from studies in yeast and transformed cell lines which proliferate continuously and therefore constitutively express active hypusinated eIF5a. In comparison, little has been reported on how eIF5a is regulated in primary mammalian cells. It was interesting therefore, that in primary T cells that transited between a naïve and an activated phenotype, a significant regulation of eIF5a activity occurred post-translationally. Naïve mouse CD8⁺ T cells have abundant eIF5a protein, but less hypusine modification compared to that found in activated T cells. Upon activation, eIF5a becomes hypusinated co-incident with the transcriptional up-regulation of DHPS and DOHH. Using CRISPR knockout of the modifying enzymes or eIF5a itself, after the cells were activated, we confirmed that hypusination was important for eIF5a function in T cells. The phenotype of the eIF5a knockout was more extreme than that of DHPS or DOHH knockouts suggesting unmodified eIF5a has some residual activity, although we cannot completely exclude that trace amounts of residual hypusinated protein remained after 4d. The deoxyhypusinated intermediate form of eIF5a had clearly detectable, albeit reduced functionality in T cells, as knockout of DOHH, which catalyses the last irreversible step in the conversion of lysine to hypusine, had much less impact than the removal of DHPS which is essential for the initial attachment of hypusine.

Controlling protein production by regulating polypeptide elongation is emerging as an important means of translational control beyond that exerted by the more extensively studied regulators of translation initiation[51]. Dysregulation of elongation has also been shown to contribute to diseases such as neurodegeneration and cancer[51,52]. The availability of aminoacyl-transfer RNAs and the action of key proteins such as eEF1A, eEF2 and eEF2K have been well documented in controlling the rate of polypeptide formation[53]. In contrast

the contribution of eIF5a is less well understood but it is thought to be particularly important for elongation through certain motifs in the mRNAs particularly those encoding proline-proline and other charged residues[10]. Our data support eIF5a regulating de novo translation of protein subsets enriched in PPP motifs, as previously described[16], although >50% of our identified eIF5a-regulated genes did not have these motifs in their coding sequences. Strong ribosome pausing has been observed in eIF5a-depleted yeast at non-proline, charged amino acid containing motifs such as DVG, DDG, GGT, and RDK[10]. Only DVG seemed slightly enriched in our datasets. In addition, mRNAs containing 5' TOP[54], IRES[55], or uORF[55] motifs were analysed and only proteins with 5' TOP mRNAs were slightly more synthesised in *EifSa* KO cells compared to bootstrapped controls (83 genes, median FC = 1.29, $P < 0.001$, two-tailed $t$ test). Online motif discovery tools such as MEME Suite[56] were not able to identify enriched motifs in the CDS of eIF5a-regulated genes compared to all expressed proteins. It is possible those mRNAs are differentially regulated by changes in ribosome composition[40,57]. There were alterations in nascent polypeptide abundance of ribosome associated proteins in eIF5a KO cells, which might influence prioritisation of mRNA translation through the influence of eIF5a on ribosomal protein translation rather than its elongation activity. Upregulation of ribosomal proteins was reported in mouse liver in a study using siRNA to knockdown expression of eEF2, a member of the GTP-binding elongation factor family[58]. Binding of eEF2 to ribosomes reduces the affinity of eIF5a-ribosome binding[59], suggesting a mutually exclusive or competitive relationship of these two translation factors in mRNA elongation. Proteins up-regulated following eEF2 knockdown were primarily ribosomal proteins as these continued to be translated relative to a background of suppressed translation. There were some similarities between the nascent proteome analysis from *EifSa* KO T cells and eEF2 knockdown cells, such as up-regulation of the translation factor eEF1b and down-regulation of eIF5, however, many more proteins were up- or down-regulated specifically in each dataset.

T cells differentiate to effector cells which secrete cytokines in a temporally and spatially regulated manner. Cytokine production in immune cells is controlled both transcriptionally and translationally, with the latter involving both miRNAs[60] and RNA-binding proteins[4,61]. We found eIF5a to be particularly important for the production of IFNγ, an important effector cytokine, both for IFNγ translation and indirectly through production of key transcription factors such as IRF4, TBET and EOMES. There is an interesting correlation between the decline of cytokine production, T cell proliferation[27] and memory T cell formation[19] observed in ageing and an age-related decline in spermidine availability and eIF5a protein abundance in lymphocytes[29,30]. Dietary spermidine supplementation in old mice has been shown to promote and improve T memory cell formation[19] and B cell antibody production[30] supporting an active contribution of eIF5a in maintaining T cell functionality in ageing, previously proposed to act via maintenance of autophagy[29]. Our data detected reduced autophagic flux in *EifSa* KO but not in *Dhps* and *Dohh* KO cells, despite decreased production of IFNγ in all KOs, suggesting a degree of autophagy-independent effect of eIF5a on IFNγ regulation, possibly via the rate of mRNA translation.

Our study looked at the influence of eIF5a on CD8⁺ T cells pre-activated prior to CRISPR gene deletion in order to identify eIF5a translationally regulated proteins in acutely activated T cells and avoid any confounders associated with impairment of lineage differentiation. In this setting, loss of eIF5a function resulted in death of the knockout population over a matter of days both in vitro and in vivo. A recent study examining polyamine metabolism with respect to CD4⁺ T cells differentiation showed that either T cell-specific loss of ornithine decarboxylase, essential for spermidine production, or loss of DOHH strongly biased differentiation of naïve CD4⁺ T cells to excessive IFNγ production[62]. This effect was manifest at the level of chromatin

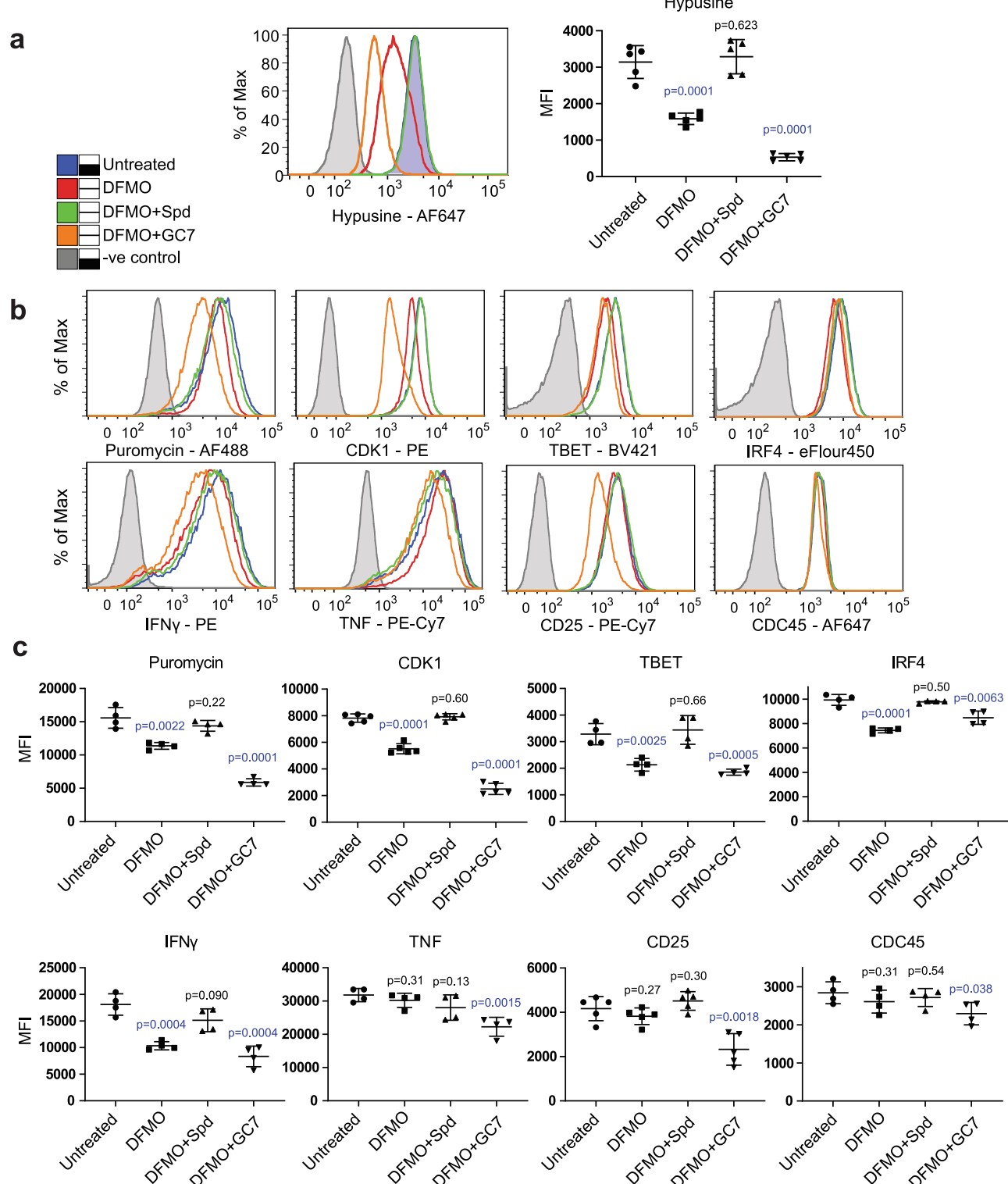

remodelling which is important during naïve CD4+ T cell differentiation and suggests eIF5a may be involved in differentiation, which was not addressed here.

GC7 is in consideration as a treatment for several cancers due to its anti-proliferative activity[63] and ability to induce cell death[64]. In our experiments GC7 produced a very different cellular phenotype from *Dhps* knockout, as was shown for a fibrosarcoma cell line[65]. GC7 competes with spermidine for binding to DHPS and has been widely used as an inhibitor for functional maturation of eIF5a. As spermidine has functions independent of eIF5a, such as inhibition of Ep300

activity, it is also likely that GC7 has comparable effects. In our LC-MS dataset, GC7 treatment resulted in 2617 down-regulated nascent proteins, while eIF5a KO decreased 234 proteins. These observations indicate a substantial inhibitory effect of GC7 independently of eIF5a. Furthermore, proteins down-regulated exclusively after GC7 treatment were not enriched in polyproline motifs, and they clustered significantly to GO terms associated with ribosome constituents, suggesting GC7 reduces cellular translation in a more global manner than eIF5a. In contrast, eIF5a knockout resulted in increased rather than decreased abundance of several ribosomal proteins.

**Fig. 7 | Restricting availability of polyamine spermidine to cells decreased eIF5a hypusination and some but not all eIF5a-regulated genes. a** Inhibition of spermidine biosynthesis by DFMO reduces hypusine staining which is rescued by exogenous supplement of spermidine or further impaired by treatment with GC7. MFIs of hypusine from 5 independent experiments are plotted on the right (*P* values were calculated using unpaired two-sided *t* test). Centre line and error bars represent mean value ±SD between independent experiments. Lines: grey shaded, -ve staining control; blue shaded, untreated; red, DFMO; green, DFMO + Spermidine; orange, DFMO + GC7. **b** Samples from (**a**) were stained for cytokines (IFNγ and TNF), transcription factors (IRF4 and TBET), cell cycle regulators (CDK1 and CDC45), CD25 (IL2 receptor α), and puromycin. **c** Pooled data of MFIs (from 4

independent experiments for Puromycin, TBET, IRF4, IFNγ, TNF and CDC45; 5 for CDK1 and CD25) were shown in the lower panel. Statistical tests were performed against untreated cells (*P* values were calculated using unpaired two-sided *t* test). Centre line and error bars represent mean value and ±SD between independent experiments. *P* value colours: blue, significant; black, non-significant. **a**, **b** OT-1 cells were stimulated with SIIINFEKL peptide for 2d and cultured in IL-2-supplemented media for a further 2d before culture with IL-2 only (Untreated), or IL-2 + 0.5 mM DFMO, DFMO + 10 μM spermidine, or DFMO + 10 μM GC7 for another 2d. Cells were restimulated with SIINFEKL peptide + Brefeldin A for the final 4 h and labelled with puromycin for 10 min before harvesting for intracellular staining and flow cytometry.

Our data support the importance of eIF5a in regulating immune cell function. Targeting eIF5a hypusination has been shown to have anti-inflammatory effects in macrophages[66] and to regulate macrophage mitochondrial respiration and activation[67] Our current study also indicates eIF5a is important for regulating proliferation and some cytokine production in T cells both through directly regulating translation of subsets of mRNAs and indirectly by influencing the production of key transcription factors. Together these studies lend weight to the existence of eIF5a directed RNA regulons important for modulating immune responses.

## Methods

### Ethics Statement

This study was approved by the Ethical Review Body at the School of Biological Sciences, University of Edinburgh. All animal experiments were approved by the University of Edinburgh Bioresearch and Veterinary Services Ethical Review body and the United Kingdom Home office under project licence P38881828 to R.Z.

### Mice

Mice expressing the OT-1 TCR transgene (C57BL/6-Tg(TcraTcrb) 1100Mjb/J) backcrossed to the Rag-1KO (B6.129S7-*Rag1*[tm1Mom]/J) background and containing congenic alleles for CD45.2 or CD45.1 were used as LN donors for CD8[+] T cells in all experiments. Mice were between 5 and 12 weeks of age and sexes were randomised for tissue donors. WT (CD45.1xCD45.2) C57BL/6 J (Charles River) mice were used as recipients for in vivo experiments. All mice were bred and housed in individually ventilated cages under specific pathogen-free conditions at the University of Edinburgh Bioresearch and Veterinary Services (BVS) facilities. Mice were maintained under a 12 h light/12 h dark cycle with ad libitum access to food and water at a temperature of 19–24 ˚C and humidity of 45–65%.

### Cell culture and stimulation

Lymph node (LN) OT-I T cells were obtained by passing tissue through 70 μM mesh filters and were cultured in IMDM medium (Sigma-Aldrich I3390) supplemented with 10% FCS, L-glutamine, 100U/ml penicillin, 100U/ml streptomycin and 50 μM 2β-ME. 100 nM SIINFEKL peptide (Cambridge Peptides, custom synthesised) was added to culture media and the cells were incubated at 37 °C for the duration specified in each experiment. On day 2 activated cells were washed and resuspended in media supplemented with 20 ng/mL recombinant human IL-2 (PeproTech AF-200-02) for the remainder of the culture period. For GC7 treatment for transcriptomic and nascent proteomic profiling, cells were grown in the presence of 10 μM GC7 (Merck Chemicals 259545) for the final 2 days. For mock treatment, an equal volume of 10 mM acetic acid vehicle was added to culture in place of GC7. For DFMO experiment, activated cells expanded for 2 days in IL-2 were cultured for an additional 2 days in fresh IL-2, or IL-2 supplemented with 0.5 mM DFMO (Bio-Techne 2761/50), DFMO + 10 μM spermidine (Sigma-Aldrich S0266), or DFMO + 10 μM GC7.

For re-stimulation of cytotoxic T cells, 100 nM SIINFEKL peptide together with 5 μg/mL Brefeldin A (Cambridge Bioscience 11861-25 mg-

CAY) to prevent cytokine secretion, were added to the culture for the final 4 h before harvest. For puromycin incorporation, puromycin was added to culture media in a final concentration of 10 μg/mL for the final 10 min before harvest.

### CRISPR-Cas9 transfection

The following guide RNA sequences were purchased from Integrated DNA Technologies: *eIF5a* (TGCTCAGCATTACGTAAGAA and CTTCGA-GACAGGAGATGCAG), *Dhps* (ATACCTCGTGCAGCACAACA), *Dohh* (GCAGTATTCTACGGACCCAG), *Thy1* (ACAGACAAGCTGGTCAAGTG), and TRAC guide RNA (CAGGGTTCTGGATATCTGT). Cas9 ribonucleoprotein complexes were formed by mixing 2 μL of 100 μM tracrRNA and 2 μL of 100 μM crRNA in 25 μL Nuclease-free duplex buffer (Integrated DNA Technologies 11-05-01-12), incubating at 95 °C for 5 min and then mixed with 2 μL of 5 mg/mL Truecut Cas9 protein (Thermo Fisher Scientific A36499) for 10 min at 37°. The RNP complexes were made immediately before the transfection and mixed with 10[6] 2-day activated OT-1 cells. Transfections were performed with the Neon transfection system (Thermo Fisher Scientific MPK5000) following manufacturer's instructions, with the setting 1600V, 10 ms, 3 pulses. Following transfection cells were cultured in IL-2-containing media for 3 days for *eIF5a* KO, and 4 days for *Dhps* and *Dohh* KO.

### AHA-labelling and cell sorting

Cells were transferred into IL-2-containing methionine-free RPMI medium at a concentration of 5 ×10[6] cells/mL. Cells were starved of methionine for one hour then 4-Azido-L-homoalanine HCl (L-AHA) (Jena Bioscience CLK-AA005-10) was added to the media (final concentration 40 μM) and incubated for 2 h at 37 °C. For experiments with *eIF5a* CRISPR KO cells they were fixed in 2% formaldehyde (Sigma-Aldrich F8775) for 20 min and permeabilised with ice-cold 90% methanol for 1 h. eIF5a staining was done in FACS Buffer (PBS + 0.5% BSA + 0.05% Sodium Azide + 4 mM EDTA) using anti-eIF5a-C-terminus antibody (Abcam ab32407, used at 1:400) overnight followed by washing and incubation with anti-rabbit IgG AlexaFluor 647 secondary antibody (Thermo Fisher Scientific A27040, used at 1:500) for 1 h. Sorting was undertaken with a FACS Aria (BD).

### Click chemistry and preparation for LC-MS

Proteins from freshly frozen cell samples (12.5 × 10[6] of non-AHA-labelled, 32.5 × 10[6] of GC7 and 12.5 × 10[6] of Mock, 4 samples each) were extracted with CHAPS Lysis buffer from the Click Chemistry Capture Kit (Jena Bioscience CLK-1065). Input cell numbers were adjusted to give similar amount of total injected peptides, ensuring similar detection coverage. For sorted fixed cells, proteins from 10 × 10[6] cells were extracted using 310 μL of 200 mM Tris, 0.5% SDS, 200 mM NaCl, pH8.0, with brief sonication to disturb the cell pellet, followed by reverse crosslinking at 95 °C for 45 min and then cooled on ice and finally mixed with an equal volume of 16 M Urea giving a final concentration of 8 M Urea. Extracted proteins were incubated overnight with Alkyne agarose beads (Jena Bioscience CLK-1032-2), and downstream preparations were done using Click Chemistry Capture Kit following the manufacturer's guidelines, followed by desalting over

C18 tips (Fisher Scientific 10627275). GC7 and Mock cells were pooled for the generation of a high pH reverse phase fractionated peptide library. Cells were lysed in PBS, 2% SDS, cOmplete protease inhibitors (Sigma 11836170001), and 20U of Benzonase (Millipore 70746), reduced with 25 mM TCEP, and alkylated with 25 mM iodoacetamide. 500 μg protein was precipitated with 80% acetone overnight at −20 °C, washed twice with 100% acetone and once in 90% ethanol. Pellets were digested with 1:50 wt/wt trypsin in 100 mM triethylammonium bicarbonate at 37 °C overnight, then desalted over a C18 Sep-Pak cartridge (Waters). Peptides were separated over a 4.6 mm XBridge BEH C18 column (Waters) in 10 mM ammonium formate (pH 9) and eluted over a linear gradient from 2-50% acetonitrile at 1 ml/min. 72 fractions were collected and concatenated into 12 fractions and dried. Peptide samples were resuspended in 0.1% TFA, half of the click chemistry captured peptide and approximately 0.5 μg of the 12 library fractions were injected for LCMS analysis. An Ultimate 3000 RSLCnano HPLC (Dionex, Thermo Fisher Scientific) was coupled via electrospray ionisation to an Orbitrap Elite Hybrid Ion Trap-Orbitrap (Thermo Fisher Scientific). Peptides were loaded directly onto a 75 μm x 50 cm PepMap-C18 EASY-Spray LC Column (Thermo Fisher Scientific) and eluted at 250 nl/min using 0.1% formic acid (Solvent A) and 80% acetonitrile/0.1% formic acid (Solvent B). Samples were eluted over 120 min stepped linear gradient from 1% to 30% B over 95 min, then to 45% B over a further 25 min. MS1 scans were acquired at 120k resolution over 350–1700 $m/z$ and a 'lock mass' of 445.120025 m/z was used. This was followed by 20 data-dependent MS2 CID events in the ion trap at rapid resolution with a 2 Da isolation width, 5E3 target ion accumulation, 50 ms maximum fill time, a normalised collision energy of 35, a requirement of a 10k precursor intensity, and a charge of 2+ or more. Precursors within 5 ppm were dynamically excluded for 40 s.

## Systematic dataset handling

Pair-end RNA sequencing dataset reads were generated by Novogene (GC7- and Mock-treated samples, 4 biological replicates) and were trimmed of low-quality and adapter sequences using Cutadapt v2.3 and fastqc v0.11.8. Reads were aligned to the mouse genome (GRCm38.96) using STAR v2.7.1a[68] and samtools v1.9. Mapped reads were counted using featureCounts function in Rsubread v1.6.4[69] and gene annotations added with rtracklayer v1.42.2. Differential expression analysis was performed with edgeR v3.26.4[70].

Nascent proteomic data was analysed using MaxQuant version 1.6.2.6[71]. LC-MS/MS data was searched against the mouse reference proteome from UniProt derived from mouse genome GRCm38 (accessed on 5 February 2018), which contains 52,599 entries, allowing for 2 tryptic missed cleavages, allowing for variable methionine oxidation, protein N-terminal acetylation, and cysteine carbamidomethylation. The parameter "Individual peptide mass tolerance" was selected for variable precursor mass tolerances, with 0.5 Da or 20 ppm mass tolerances set for ion trap or orbitrap fragment ions, respectively. A target-decoy threshold of 1% was set for both PSM and protein false discovery rate. Match-between-runs was enabled with identification transfer within 0.7 min and a retention time alignment within 20 min window. Matching was permitted from the library parameter group, and 'from and to' the unfractionated parameter group, and second peptide search was enabled. Level of a protein was represented by the median intensity of all peptides attributed to it, and only proteins containing more than 2 unique detected peptides were deemed expressed. iBAQ values were used for analysis. All values were normalised using Vsn algorithm[32] to adjust for unequal sample loading. Since different number of input cells were used between groups to achieve similar coverage, the Vsn-normalised iBAQs were then normalised by the input cell numbers (Detailed in Supplementary Fig. 3a). Statistical tests were performed using log2-transformed normalised iBAQ values. Unpaired T test with Benjamini–Hochberg correction[72] was used to determine significantly down-regulated proteins between

GC7- and Mock-treated samples. Paired T test was used for eIF5a KO versus WT samples. Differential expression analyses were done in Excel 2016.

## Flow cytometry

Cells were labelled with LIVE/DEAD Aqua Dead Cell Stain Kit (Thermo Fisher Scientific L34966) and then fixed in Intracellular Staining Fixation Buffer (BioLegend 420801) and permeabilised in Intracellular Staining Permeabilization Wash Buffer (BioLegend 421002). Antibodies against eIF5a (EP527Y, Abcam ab32407, used at 1:400), hypusine (Hpu24, Creative Biolabs PABL-202, used at 1:200), IFNg (XMG1.2, BioLegend 505808, used at 1:400), TNF (MP6-XT22, eBioscience 25-7321-82, used at 1:400), IRF4 (3E4, eBioscience 48-9858-82, used at 1:200), TBET (4B10, BioLegend 644832, used at 1:200), EOMES (Dan11mag, Thermo Scientific 53-4875-82, used at 1:200), Granzyme B (GB11, BioLegend 515403, used at 1:200), CDK1 (A17, Abcam ab18, used at 1:200), CD25 (PC61, BioLegend 102016, used at 1:200), CD98 (RL388, eBioscience 12-0981-83, used at 1:200), PD-1 (RMP1-30, eBioscience 11-9981-82, used at 1:200), CDC45 (EPR5759, Abcam ab126762, used at 1:200) and puromycin (12D10, Millipore MABE343-AF488, used at 1:500) were used; for cell cycle analysis, fixed-permeabilised cells were stained with Hoechst 33342 (Sigma) and Ki-67 antibody (B56, BD Biosciences 556027, used at 1:10). Cells were incubated with primary antibodies for 16 h at 4 °C, washed, followed by staining with Goat anti-rabbit IgG Alexa Fluor 647 (Thermo Fisher Scientific A27040, used at 1:500) and Rat anti-mouse IgG2a PE (RMG2a-62, BioLegend 407108, used at 1:200) secondary antibody for 1 h at RT. Cells were resuspended in FACS Buffer (described in cell sorting) before acquisition. All analyses were done on singlet live events.

For quantitation of autophagic flux, cells were treated with/without 10 nM bafilomycin A1 (BafA1, Sigma Aldrich B1793) for 2 h, followed by selective permeabilisation and LC3 staining using Flow-Cellect Autophagy LC3 antibody-based assay kit (Luminex FCCH100171). Stained cells were fixed with 2% formaldehyde and resuspended in FACS Buffer before acquisition. Autophagic flux of a sample was defined as the mean fluorescence intensity of its LC3-II in BafA1-treated cells divided by that in mock-treated cells.

For cytotoxicity assay, mouse lymphoma line EL-4 cells were pulsed with SIINFEKL peptide for 1 h at 37 °C and washed with IMDM, then labelled with CellTrace Violet (CTV, Invitrogen C34557); non-pulsed EL-4 cells were labelled with CFSE (Invitrogen C34554). In 96 well U-shaped plate $9 \times 10^5$ SIINFEKL-pulsed EL4 cells were combined with $9 \times 10^5$ non-pulsed cells, and a titration of T cells starting from $5 \times 10^6$ cells and 3-fold dilutions. Cells were incubated for 4 h and acquired looking at change of ratio of CFSE:CTV labelled EL4s. A repeat experiment swapping CFSE and CTV was done.

All acquisitions were done on a MACSQuant flow cytometer (Miltenyi Biotec). Data were analysed with Flowjo version 9.

## In vivo dynamics of CRISPR KO cells

$1 \times 10^5$ Day 3 *Eif5a* or Day 4 *Dhps/Dohh* CRISPR OT-1 CD45.2 cells were combined with $1 \times 10^5$ Day 4 *Thy1* CRISPR OT-1 CD45.1 cells and $1 \times 10^6$ *Listeria monocytogenes* expressing OT-1 TCR target chicken ovalbumin (*Lm*OVA) and injected intravenously into WT C57BL/6 mice. Day 6 post-injection mice were sacrificed and their splenocytes harvested. Cells were restimulated ex vivo with SIINFEKL peptide in the presence of 5 μg/mL Brefeldin A for 4 h and then stained for CD45.1, CD45.2, eIF5a/hypusine and IFNγ.

## Protein extraction and Western blotting

Cells were lysed in RIPA Buffer (Thermo Fisher Scientific 89900) supplemented with a protease inhibitors cocktail (Sigma-Aldrich P8340). The lysates were centrifuged to remove the debris. Reducing Laemmli buffer was added to the lysates, and samples were heated to 95 °C and

separated by SDS-PAGE. Proteins were transferred to Immobilon-FL PVDF membrane (Millipore IPFL00010), and membranes were incubated in Odyssey blocking buffer (LI-COR Biosciences 927-40000) prior to incubation with the primary antibodies. Quantitative signals were generated with secondary Abs: IRDye® 680RD Goat anti-Rabbit and IRDye® 800CW Goat anti-Mouse (LI-COR Biosciences 926-68071 and 926-32210, both used at 1:5000), and visualised using the Odyssey Infrared Imaging System (LI-COR Biosciences).

The following primary Abs were used: anti-ZAP70 (clone 29, BD Biosciences 610239, used at 1:1000), DHPS (Abcam ab224134, used at 1:1000), eIF5a (EP527Y, Abcam ab32407), and hypusine (rabbit polyclonal Millipore ABS1064, used at 1:1000; rabbit monoclonal Hpu24 Creative Biolabs PABL-202, used at 1:1000).

### Transcript analyses by RT-qPCR
Total cellular RNA was extracted using Direct-zol RNA Miniprep kit (Zymo Research R2051) and reverse transcribed using Lunascript kit (New England Biolabs E3010L). qPCR was performed on a Light Cycler 480 (Roche) using Brilliant III SYBR® Master Mixes (Agilent 600883). Transcription level of each gene was normalised against the quantity of beta actin (Actb) in the sample. Oligonucleotides (custom synthesised by Integrated DNA Technologies) are listed in Supplementary Table 1.

### Quantification and statistical analysis
Prism 9 software (GraphPad) and Excel 2016 were used for statistical analyses. Unless stated otherwise, all data points represent the measurement of biological replicates. Unpaired or paired two-tailed Student's $t$ test was used for comparisons between two normally distributed datasets with equal variances unless specified. One-tailed Fisher's exact test was used for testing enrichment of amino acid trimers. In the GC7-Mock nascent proteome dataset, Benjamini–Hochberg method was used to adjust $P$ values of a family of multiple t tests with a single null hypothesis. $P$ value was used to quantify the statistical significance of the null hypothesis testing. *$p \leq 0.05$, **$p \leq 0.01$.

### Reporting summary
Further information on research design is available in the Nature Portfolio Reporting Summary linked to this article.

## Data availability
The data that support this study are available from the corresponding author upon request. Original and processed RNASeq dataset have been deposited in the Gene Expression Omnibus (GEO) database, under accession no. GSE168731. Original files for the nascent proteomic dataset are available at EBI PRIDE database, under accession no. PXD021063. Source data are provided with this paper and also available on Open Science Framework at https://doi.org/10.17605/OSF.IO/J94BY. Source data are provided with this paper.

## Code availability
Source codes are available on Open Science Framework at https://doi.org/10.17605/OSF.IO/J94BY.

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

## Acknowledgements
This research was funded by the Wellcome Trust Investigator Award WT205014/Z/16/Z to R.Z. and by the Wellcome Trust/Royal Society Sir Henry Dale Fellowship 20611/Z/17/Z to T.L. We would like to thank the following at the University of Edinburgh: Dr Martin Waterfall IIIR for cell sorting expertise, the BVS animal facility in Ashworth labs and CRM for support with animal husbandry, and the proteomics facility at the Wellcome Centre for Cell Biology for mass spec. We also thank Dr Alehandro Brenes Murillo at the University of Dundee for valuable discussion and help regarding dataset analysis.

## Author contributions
Conceptualisation: T.T. & R.Z.; Methodology: T.T., V.K. and TL; Investigation: T.T., X.Z., D.W. and V.K.; Formal analysis: T.T.; Data curation: T.T. and V.K.; Writing—original draft: T.T. and R.Z.; Writing review and editing: T.T., T.L. and R.Z.; Funding acquisition: T.L. and R.Z.; Resources: R.Z.; Supervision: R.Z.

## Competing interests
The authors declare no competing interests.
