## [Peer Review File · Nature Communications]

Translation factor eIF5a is essential for IFN γ production and cell cycle regulation in primary CD8 T lymphocytesREVIEWER COMMENTS

Reviewer #1 (Remarks to the Author):

The manuscript by Tan et al shows how the translation factor eIF5a (and its regulation through hypusination) regulates the production of specific proteins in T cells during activation. The authors here dissect the role of hypusinated EIF5a in T cells and clearly demonstrate that it affects the production of cytokines (in particular IFN γ), and that several transcription factors are involved. Proteomics analysis together with transcriptomics data from GC7 treated cells revealed target RNAs for hypusinated EIF5a. The authors also showed that possibly unhypusinated EIF5a has translation activity- but to which targets is not resolved. They also show that chemical inhibition and deletion of enzymes for hypusination does not fully recapitulate the effects of hypusinated EIF5a, which points to other roles of the enzymes. The possible mechanisms are however only speculated about. I find this manuscript interesting and surely of use to the community, as it for the first time systematically studies the role of EIF5a in translation in CD8 T cells. However, I also have several points that require attention of the authors and that should be addressed.

Figure 1C: I am surprised that the induction of EIF5a (and its hypusination) is bi- or even tri-modal rather than a gradual increase. How should one interpret this? And how do the authors interpret the higher increase on hypusination? Is this really alterations of EIF5a already present or does hypusination only occur on de novo synthesized EIF5a? One could also interpret these data as an artefact of the antibodies, or as hypusination of other, yet to be defined proteins? Please see my point on this below.

Figure 2A; how come that HOECHST staining from GC7 treated cells looks so different to the ko panels? Is it from a different experiment? Then it should be marked in a different subpanel. Or is this observed difference due to the drug? And why do the authors use different methods to look at the effect? I.e. GC7 effects on S phase while the others is G1? This is confusing. Also because the depicted histograms suggest that populations in S and G2 phase seem to be most affected by eIF5a/Dhps/Dohh KOs.

Figure 2B: I appreciate the use of non-edited cells as an internal control in the same mix of cells as the authors do. However, I miss the control of the Thy1 deleted T cells. Gene-editing itself may hamper or delay the capacity to proliferate and/or to produce cytokines, and this effect is not captured by comparing it to non-edited T cells in the same culture. So the Thy1 deletion needs to be added in these essays.

Also For qRT-PCR, (Figure 2 and 5) cell handling (FACS sorting) can substantially influence gene expression. Therefore the authors should include Thy1 edited cells for a fair comparison. How does Puromycin incorporation look like for GC7 treated cells?

Line 169: =restimulated with peptide': please be more specific with which peptide and how much (also in the figure legends this info is missing. Critical for people to be able to reproduce and easily judge data.

At page 2/3 how much does the ko of This and Dohh influence the hypusination? Is it a full block, or a substantial reduction? In other words: is the residual translation efficiency that is measured a result of non-hypusinated EIF5a, or is it from the few hypusinated EIF5a that is still present? A Westernblot of FACS sorted CRISPRed cells would shed light on this question.

Figure 3: I am somewhat surprised on the data analysis pipeline the authors used for the proteomics data. iBAQ values are useful when comparing abundance of protein relative to each other (in the same sample). However, for the analysis pipeline employed here LFQ is the prime choice to compared data between samples. In fact, what mostly is done is analysing LFQ values in the DEP package or perseus. Those packages also allow imputation which should be a more robust system for the analysis pipeline.

In fact am not aware that iBAQ is used for statistics. Can the authors refer to a paper that uses iBAQ for this purpose? And provide some reference for the normalizations that they do?

For the imputation, the authors should provide a graph with missing values, because they give provisional values to 0s, which could in fact bias their findings. It may also be the reason why some groups show so high LFCs in their analysis, something that is quite unusual in this type of data.

Also, according to the scheme presented in the supplemental figure a protein is considered detected when measured in 4 out of 16 samples. This phrasing is not clear to me. Does it mean that a peptide is considered also detected when it is present in only 1 out of 4 replicates only? A peptide should be robustly measured in the replicates and be present in at least 3 (or at least 2) out of 4 replicates prior to being considered detected.

Is this low threshold (that i suspect) also the reason why the number of enriched proteins is so high (>5000) after only 2h of labeling? I ask this because in most studies the enriched proteins are rather in the order of hundreds, and not thousands. Please make sure that pipelines are well described and according to the standards.

I would like to emphasize that these issues of data analysis are relatively easy to fix and should 'clean up' the data, and strengthen the conclusions that can be drawn.

Line 271-277/Figure 4A:

The authors should provide some QC data on the RNA seq data which now are missing, such as a volcano plot in the supplemental data would already help, without already focussing on the RNAs that are differentially regulated. I am also not sure how the cut-offs are chosen to consider an RNA expression not altered- please also include this information. Tables of only names as provided now are difficult to follow for the reader.

On a side note: While I think that the approach used by Tan et al to perform RNAseq on GC7 treated cells is valid, it is the second best, as also mentioned by the authors. I would like to draw the attention to the reviewers that in contrast to what they state, the issue to perform RNA-seq on FA-fixed cells has been solved by several labs:<https://doi.org/10.1371/journal.pone.0240769>, <https://pubmed.ncbi.nlm.nih.gov/32161126>.

I also want to encourage the authors to show values in the tables that are provided as names are useless to the reader, if one cannot put numbers of enrichments to it and learn which one are somewhat enriched and which are highly enriched in the presence/absence of EIF5a.

Figure 5a: While the Flow cytometry plots for TBET and CDK1 are very convincing, I am much less convinced on the data on IRF4. The ratio of ko versus WT is off (more 5-10% ko, instead of 50% in Figure 2), and might thus skew the results. It would be good to have other examples where the ko is more efficient (as shown in Fig 2 where 50% was achieved), comparable to what is shown for TBET and CDK1 where the ratio is less off. However, what is 'above threshold' for CDK1 expression? Also that is difficult to interpret as a reader.

Figure legends should make figures self-explanatory, but now miss quite some details that allow the usage of them as such, and critical information can only be found in methods and/or in the results section. Please be more specific in your experimental description.

While I found no other publications on hypusination other than on EIF5a, I am struck that the papers I checked- including this manuscript- do not take into account the possibility that hypusination could in principle also occur at other yet to be identified proteins, which could also be the source of some of the findings. While I don't think it is a main point, it would render the discussion more balanced and accurate. Similarly, the statement of the authors in line 448-449 could also be due to autophagy regulation. It should also be rephrased.

Minor points:

Methods: authors failed to mention what MOCK treatment entails in cell culture and stimulation. Please specify.

Line 356: please refer to Figure 6B here.

Line 169: =restimulated with peptide': please be more specific with which peptide and how much (also in the figure legends this info is missing. Critical for people to be able to reproduce and easily judge data.

The title 'essential role of EIF5a on cytokine production' – is a bit misleading. Even though the authors show substantial effects on IFN γ production between 5-30% of the cells produce IFN γ (Fig 2B). in addition, the effect on TNF production is limited, and other cytokines are not measured. Therefore, the title should be more precise on what the authors report.

Reviewer #2 (Remarks to the Author):

There is a growing appreciation for research aimed at increasing our current understanding of the role of mRNA translation in immune cells. In this manuscript the authors focus on describing the role of eIF5a and its hypusination in T cells. They describe a role for eIF5a in regulating translation of specific subsets of mRNA during T cell activation. Studies include the use of CRISPR knockouts of eIF5a, DHPS and DOHH (2 enzymes required for hypusination of eIF5a). Chemical inhibitors such as the deoxyhypusine synthase inhibitor, GC7 are also used.

Overall the work is largely descriptive in nature, and to warrant publication in Nature Communications should include additional functional studies for the role of eIF5a in an in vivo context. All the studies shown in this manuscript focused on T cells which were artificially activated ex vivo. While yes, we now have a description of transcriptional regulators which may be under the control of eIF5a translation, there are no studies linking this to T cell function. The authors focus on a few cytokines, such as IFN-gamma and TNF-alpha, but no in vivo work has been done to support that T cells are less fit in mounting an immune response when eIF5a or enzymes which modify its hypusination are lost. As such, enthusiasm for publication of this manuscript in its current form is low. Some specific comments are listed below:

- In figure 2, the authors should show a kinetic response in their T cell stimulation assays, that is, over time and not just the one time point shown in the figure. As it stands, the data shown for each condition assessed is a snapshot at a differing time points, 2 days for GC7, 3 days for eIF5a, 4 days for DHPS and DOHH, perhaps confounding the interpretation of the results. The authors find that the stimulated T cells (modified genetically or pharmacologically) failed to produce effector cytokines, but what about other important readouts, such as the expression of inhibitory immune-receptors which are linked with T cell exhaustion. Similarly, why not assess T cell activation state or cytolytic function of the modified T cells?
- Related to the data shown in figure 2, the authors make statements about translation being affected or not without showing assays that test the impact of their genetic or chemical modifications on translation. Caution should be used. For example, "the loss of IFN γ production was selective rather than reflecting a general shut down in cytokine translation, as all the targeted cells showed a much smaller, albeit significant, reduction in TNF α production." and "Taken together, these results indicate that lack of fully matured eIF5a does not impair TNF α mRNA transcription and has a very mild effect on its translation. By contrast, mature eIF5a is required both for transcription and translation of IFN γ ." No assays were used to support that translation of IFN γ or TNF α were affected.
- It is important to assay the impact on autophagy in the experiments wherein the authors use spermidine, modulate eIF5A, and use GC7. As noted by the authors in the text, but not

experimentally, spermidine can promote autophagy. Studies have shown that spermidine can promote autophagy via eIF5a, in other contexts (PMID: 31474573), which promotes the synthesis of the autophagy transcriptional regulator TFEB. How do the authors explain not seeing autophagy as an identified pathway in their list of translationally regulated genes affected by GC7 and eIF5aKO (Figure 4). Can a difference in autophagy status (autophagic flux) be detected when the eIF5a axis is genetically or pharmacologically modified, and thus help to explain the observed discrepancies in phenotypes of GC7-treated cells and the cells lacking eIF5a or DHPS?

Reviewer #3 (Remarks to the Author):

The manuscript by Tan et al investigates the role of translation factor eIF5a in murine primary OT-I T cells. eIF5a is amongst the top 20 most abundant proteins expressed in murine cytotoxic CD8 T cells according to previous work (Hukelmann et al NI 2015) and there is evidence that protein translation plays a key role in the regulation of protein expression in activated and memory T cells (eg. Salerno et al NI 2018; Wolf et al NI 2020).

The authors used CRISPR-Cas9 in OT-I cells to target eIF5a as well as the enzymes (DHPS and DOHH) involved in its hypusination, a post-translational modification that is apparently necessary for its activity in facilitating peptide bond formation. The authors demonstrate substantial reductions in global protein synthesis, T cell proliferation and IFN γ production as well as a minor diminution in TNF α . The effects produced by targeting eIF5a were partially replicated by Dhps deletion and to a lesser extent in Dohh KO cells. LC-MS was then used to profile the T cell proteome, which identified several proteins involved in cell proliferation as well as transcription factors such as Irf4 and Tbx21 that are known to play roles in CD8 T cell activation and differentiation all of which were reduced in eIF5a KO T cells.

It appears that Infg mRNA is also reduced in eIF5a KO cells (Fig. 2C) and there may be similar reductions in Irf4 and Cdk1 mRNA (Fig. 5B), although the data for Irf4 and Cdk1 is highly variable and requires additional data points. It is also not clear why the Gene/Actb ratios for Irf4 and Cdk1 are very different between control samples (Thy1 KO and WT) in Fig 5B. There is a similar discrepancy in Fig. 2C for detection of TNF α RNA (compare Thy 1KO and WT ratio). If eIF5a activity has an impact on transcription and/or stability of these transcripts, what are the mechanism responsible for this effect? It would also have been interesting to evaluate the effect of eIF5a on translation of repressed mRNAs eg. 5'TOP containing transcripts that are translated following mTORC1 signalling.

Finally, whilst the data clearly point to defects in cell proliferation and IFN γ expression, the findings do not put the role of eIF5a in context of other proteins of the translational machinery.

We thank the reviewers for their comments and have substantially revised the manuscript in the light of these (point by point rebuttal below) and have included a number of new Figs with new data, summarised below:

- Fig 1c, gating adjusted to reduce second peak
- Fig 2b, western blot of sorted eIF5a negative cells
- Fig 2c, autophagy
- Fig 3a & Fig S1, inclusion of Thy1KO controls and time course for FACS analysis
- Fig S1 in vivo responses of targeted cells
- Fig 3c, inclusion of mock control and eIF5aKO in qPCR analysis
- Fig 3d, cytotoxicity assay
- Fig 6 & FigS4 more comprehensive validation of targets with qPCR and time course for FACS analysis
- Fig S3c RNA-seq raw counts and volcano plot

We hope that the revised manuscript will be of interest to your readers and acceptable for publication.

Reviewer #1 (Remarks to the Author):

The manuscript by Tan et al shows how the translation factor eIF5a (and its regulation through hypusination) regulates the production of specific proteins in T cells during activation. The authors here dissect the role of hypusinated EIF5a in T cells and clearly demonstrate that it affects the production of cytokines (in particular IFN γ), and that several transcription factors are involved. Proteomics analysis together with transcriptomics data from GC7 treated cells revealed target RNAs for hypusinated EIF5a. The authors also showed that possibly unhyposinated EIF5a has translation activity- but to which targets is not resolved. They also show that chemical inhibition and deletion of enzymes for hypusination does not fully recapitulate the effects of hypusinated EIF5a, which points to other roles of the enzymes. The possible mechanisms are however only speculated about. I find this manuscript interesting and surely of use to the community, as it for the first time systematically studies the role of EIF5a in translation in CD8 T cells. However, I also have several points that require attention of the authors and that should be addressed.

We are pleased the review found our data interesting and hope that we have adequately addressed their concerns below.

Figure 1C: I am surprised that the induction of EIF5a (and its hypusination) is bi- or even tri-modal rather than a gradual increase. How should one interpret this? And how do the authors interpret the higher increase on hypusination?

We think the second peak is an artifact resulting from integrin activation upon stimulation causing aggregates of cells and at the early time points it can be quite difficult to electronically gate these aggregates out using our normal single cell gating strategies. In support, we have applied tighter FSC/SSC gating and the second peak reduces but it does not disappear completely and further tightening of the gates would remove too many cells. This is not a problem at the later time points. We have added a comment explaining this in the text in page 6 and updated Fig 1c.

Is this really alterations of EIF5a already present or does hypusination only occur on de novo synthesized EIF5a? One could also interpret these data as an artefact of the antibodies, or as hypusination of other, yet to be defined proteins? Please see my point on this below.

We think it is most likely that both newly synthesised and pre-existing pools of eIF5a become hypusinated, as the relative increase in hypusination is greater than that of eIF5a protein upon activation. It is unlikely that this is an artifact of the antibodies, as the fold increase is determined independently with respect to each antibody staining at the 0h time point. There are several early studies notably Cooper et al., 1982 which convincingly show that only one protein in the eukaryote proteome is hypusinated and that is eIF5a.

Figure 2A; how come that HOECHST staining from GC7 treated cells looks so different to the ko panels? Is it from a different experiment? Then it should be marked in a different subpanel.

Yes, the Hoechst staining from GC7 and CRISPR cells were done in different experiments acquired using different PMT settings on the FACS machine. A line is now drawn separating the two and this has been made clear in the figure legend.

Or is this observed difference due to the drug? And why do the authors use different methods to look at the effect? I.e. GC7 effects on S phase while the others is G1? This is confusing. Also because the depicted histograms suggest that populations in S and G2 phase seem to be most affected by eIF5a/Dhps/Dohh KOs.

We apologise for the confusion. The drug does act differently to the KOs causing S phase accumulation, while Eif5a and Dhps CRISPR caused G1 accumulation. We have now represented this as stacked bars in Fig 2 which compares all samples for each phase of the cell cycle and makes this point more clearly.

Figure 2B: I appreciate the use of non-edited cells as an internal control in the same mix of cells as the authors do. However, I miss the control of the Thy1 deleted T cells. Gene-editing itself may hamper or delay the capacity to proliferate and/or to produce cytokines, and this effect is not captured by comparing it to non-edited T cells in the same culture. So the Thy1 deletion needs to be added in these essays.

Also For qRT-PCR, (Figure 2 and 5) cell handling (FACS sorting) can substantially influence gene expression. Therefore the authors should include Thy1 edited cells for a fair comparison.

The Thy1 KO controls have now been added throughout to the histogram overlays and the use of a two CRISPR guides increased the KO efficiency of the eIF5a cells to >80%, so comparable with the Dohh and Dhps samples, and we no longer needed to sort them before RNA extraction removing issues with needing to sort cells.

How does Puromycin incorporation look like for GC7 treated cells?

The effect of GC7 on puromycin incorporation is now shown in the new Fig 3a. The reduction in puromycin incorporation is not as severe as KO of eIF5a and more similar to Dhps and Dohh KO

Line 169: '=restimulated with peptide': please be more specific with which peptide and how much (also in the figure legends this info is missing. Critical for people to be able to reproduce and easily judge data.

Apologies for the omission these details have now been included

At page 2/3 how much does the ko of This and Dohh influence the hypusination? Is it a full block, or a substantial reduction? In other words: is the residual translation efficiency that is measured a result of non-hypusinated EIF5a, or is it from the few hypusinated EIF5a that is still present? A Westernblot of FACS sorted CRISPRed cells would shed light on this question.

The data suggest that it is a full block, certainly by d3 and d4 once any residual hypusinated protein has been turned over, as demonstrated by the time course FACS data in Figure S1 and as expected from removal of the key modifying enzymes. As FACS gives sensitivity at a single cell level in contrast

to WB which measures protein abundance on a population level, we think FACS is a more sensitive readout. We draw the reviewers attention to new Fig 2b WB and Fig 2a FACS which illustrates for eIF5a that sorted FACS negative cells have no detectable band by WB, illustrating good concordance between the two techniques. However, whether there is a small amount of residual hypusinated eIF5a remaining which is below these detection levels neither FACS nor WB can completely exclude.

Figure 3: I am somewhat surprised on the data analysis pipeline the authors used for the proteomics data. iBAQ values are useful when comparing abundance of protein relative to each other (in the same sample). However, for the analysis pipeline employed here LFQ is the prime choice to compared data between samples. In fact, what mostly is done is analysing LFQ values in the DEP package or perseus. Those packages also allow imputation which should be a more robust system for the analysis pipeline. In fact am not aware that iBAQ is used for statistics. Can the authors refer to a paper that uses iBAQ for this purpose? And provide some reference for the normalizations that they do?

The original paper describing LFQ normalization (PMC4159666) states that "MaxLFQ has the prerequisite that a majority population of proteins exists that is not changing between the samples". We expected that modulation of eIF5a would cause a major change in the nascent proteome. Therefore, we decided to use iBAQ, which does not have this prerequisite assumption. iBAQ has been used to quantitate proteins previously (see e.g., 10.1038/nature10098 and PMC3261714). We did not aim to perform absolute quantitation, so we did not apply a scaling factor to the iBAQ intensities as done in this ref: 10.1038/nature10098. Clearly GC7 and eIF5a KO significantly perturbs the nascent proteome (see Fig. 3a e.g.), so we believe the use of iBAQ was justified.

For the imputation, the authors should provide a graph with missing values, because they give provisional values to 0s, which could in fact bias their findings. It may also be the reason why some groups show so high LFCs in their analysis, something that is quite unusual in this type of data. Also, according to the scheme presented in the supplemental figure a protein is considered detected when measured in 4 out of 16 samples. This phrasing is not clear to me. Does it mean that a peptide is considered also detected when it is present in only 1 out of 4 replicates only? A peptide should be robustly measured in the replicates and be present in at least 3 (or at least 2) out of 4 replicates prior to being considered detected.

A new analysis workflow is used in our revision, taking only proteins detected in 3 out of 4 replicates in all conditions, and then all blank values replaced by the average iBAQ of the other 3 replicates, thereby eliminating all blank values.

Missing values in the raw data are now shown in Fig S2b, imputed with a Log2 value of 10.

Is this low threshold (that i suspect) also the reason why the number of enriched proteins is so high (>5000) after only 2h of labeling? I ask this because in most studies the enriched proteins are rather in the order of hundreds, and not thousands. Please make sure that pipelines are well described and according to the standards.

The analysis workflow for the nascent proteomic dataset is shown in Fig S2a. After the revised expression cut-off, 3246 proteins are used in DE analysis. We found the AHA-incorporation method allows detection of approx. 3-fold more nascent proteins in MS compared to OPP-labelling, this may explain the improved detection from previous studies. We also found more proteins detected in T cells compared to HEK cells prepared in the same method (3000s vs 1000s).

I would like to emphasize that these issues of data analysis are relatively easy to fix and should 'clean up' the data, and strengthen the conclusions that can be drawn.

We agree and thank the reviewer for their helpful suggestions

Line 271-277/Figure 4A:

The authors should provide some QC data on the RNA seq data which now are missing, such as a volcano plot in the supplemental data would already help, without already focussing on the RNAs that are differentially regulated. I am also not sure how the cut-offs are chosen to consider an RNA expression not altered- please also include this information. Tables of only names as provided now are difficult to follow for the reader.

A violin plot for RNASeq raw counts and a volcano plot for RNASeq DE analysis are now shown in Fig S3c. Selection cut-offs are described in the figure legend.

On a side note: While I think that the approach used by Tan et al to perform RNAseq on GC7 treated cells is valid, it is the second best, as also mentioned by the authors. I would like to draw the attention to the reviewers that in contrast to what they state, the issue to perform RNA-seq on FA-fixed cells has been solved by several labs:

<https://doi.org/10.1371/journal.pone.0240769>, <https://pubmed.ncbi.nlm.nih.gov/32161126>.

We thank the reviewer for this suggestion and we tried it out, however the technique did not work for us to get good enough quality cDNA from our T cells. Instead we used 2 guides for targeting eIF5a and improved the KO efficiency to 80% comparable to the Dohh and Dhps KOs, so we extracted RNA from the unfixed population. Additional qPCR data from this material is provided in Fig 3, Fig 6 and Fig S4.

I also want to encourage the authors to show values in the tables that are provided as names are useless to the reader, if one cannot put numbers of enrichments to it and learn which one are somewhat enriched and which are highly enriched in the presence/absence of EIF5a. *Values are now incorporated in our new table in Fig 5.*

Figure 5a: While the Flow cytometry plots for TBET and CDK1 are very convincing, I am much less convinced on the data on IRF4. The ratio of ko versus WT is off (more 5-10% ko, instead of 50% in Figure 2), and might thus skew the results. It would be good to have other examples where the ko is more efficient (as shown in Fig 2 where 50% was achieved), comparable to what is shown for TBET and CDK1 where the ratio is less off. However, what is 'above threshold for CDK1 expression? Also that is difficult to interpret as a reader.

In new Fig 6, IRF4 is now represented by an experiment with comparable Eif5a KO efficiency to TBET and CDK1. CDK1 is now quantified using MFI as for the other proteins.

Figure legends should make figures self-explanatory, but now miss quite some details that allow the usage of them as such, and critical information can only be found in methods and/or in the results section. Please be more specific in your experimental description.

All figure legends have been revised and should now contain all necessary information.

While I found no other publications on hypusination other than on EIF5a, I am struck that the papers I checked- including this manuscript- do not take into account the possibility that hypusination could in principle also occur at other yet to be identified proteins, which could also be the source of some of the findings. While I don't think it is a main point, it would render the discussion more balanced and accurate. Similarly, the statement of the authors in line 448-449 could also be due to autophagy regulation. It should also be rephrased.

That eIF5a is the only hypusinated protein was most clearly demonstrated by the publications by Cooper et al who used ³H-Spermidine to label cells and on 2D gels saw only a single spot. Similarly in yeast only a single protein containing hypusine is detected (PMID: 8347280) so remarkably eIF5a

seems to be the sole hypusinated protein in eukaryotes. We have made this evidence clearer in the text.

Minor points:

Methods: authors failed to mention what MOCK treatment entails in cell culture and stimulation. Please specify.

A description of mock treatment is now included under “Cell culture and stimulation” in Methods.

Line 356: please refer to Figure 6B here.

Updated with the new figure reference.

Line 169: ‘restimulated with peptide’: please be more specific with which peptide and how much (also in the figure legends this info is missing. Critical for people to be able to reproduce and easily judge data.

Done

The title ‘essential role of EIF5a on cytokine production’ – is a bit misleading. Even though the authors show substantial effects on IFN γ production between 5-30% of the cells produce IFN γ (Fig 2B). in addition, the effect on TNF production is limited, and other cytokines are not measured. Therefore, the title should be more precise on what the authors report.

We have made the title more specific

Reviewer #2 (Remarks to the Author):

There is a growing appreciation for research aimed at increasing our current understanding of the role of mRNA translation in immune cells. In this manuscript the authors focus on describing the role of eIF5a and its hypusination in T cells. They describe a role for eIF5a in regulating translation of specific subsets of mRNA during T cell activation. Studies include the use of CRISPR knockouts of eIF5a, DHPS and DOHH (2 enzymes required for hypusination of eIF5a). Chemical inhibitors such as the deoxyhypusine synthase inhibitor, GC7 are also used.

Overall the work is largely descriptive in nature, and to warrant publication in Nature Communications should include additional functional studies for the role of eIF5a in an in vivo context. All the studies shown in this manuscript focused on T cells which were artificially activated ex vivo. While yes, we now have a description of transcriptional regulators which may be under the control of eIF5a translation, there are no studies linking this to T cell function. The authors focus on a few cytokines, such as IFN- γ and TNF- α , but no in vivo work has been done to support that T cells are less fit in mounting an immune response when eIF5a or enzymes which modify its hypusination are lost. As such, enthusiasm for publication of this manuscript in its current form is low. Some specific comments are listed below:

Our main aim was to provide a comprehensive report on the targets of this important translation factor in a primary cell population, as opposed to the more usually studied cell lines which have different metabolic and growth requirements, which we report through analysis of the nascent proteome. Given KO T cells fail to thrive and recovery even in vitro after 4d is significantly reduced (Fig 2a) we were less confident we could examine long term function of the KO cells. However, in light of the reviewers concerns we have measured killing activity in vitro (new Fig 3d). We also tried to use the KO cells in an in vivo response but as we suspected, we were unable to recover KO cells after challenge with Listeria OVA (Fig S1b).

- In figure 2, the authors should show a kinetic response in their T cell stimulation assays, that is, over time and not just the one time point shown in the figure. As it stands, the data shown for each condition assessed is a snapshot at a differing time points, 2 days for GC7, 3 days for eIF5a, 4 days for DHPS and DOHH, perhaps confounding the interpretation of the results.

Time course analysis has now been performed and presented in Figs S1 and S4. For many of the molecules we are following eg cytokines their expression increases following restimulation and interestingly IFN γ does not upregulate in the KO cells

The authors find that the stimulated T cells (modified genetically or pharmacologically) failed to produce effector cytokines, but what about other important readouts, such as the expression of inhibitory immune-receptors which are linked with T cell exhaustion. Similarly, why not assess T cell activation state or cytolytic function of the modified T cells?

At the time we look at the CTL after 5-6d in culture they do not show much of an exhausted phenotype. However, we have now included FACS analyses in our time courses of PD1, which upregulates as an activation marker, and GzB for its relevance to CTL killing, which are unchanged (Fig S1). Rather than monitor numerous individual proteins by FACS analyses we chose instead to look at the global proteome changes by mass spec of nascent proteins in the KO cells which we felt was unbiased, more informative and comprehensive. Although we focus on a limited number of proteins of interest in the manuscript we present the entire nascent proteomes in Table S1 which we hope will be a useful resource for readers.

- Related to the data shown in figure 2, the authors make statements about translation being affected or not without showing assays that test the impact of their genetic or chemical modifications on translation. Caution should be used. For example, “the loss of IFN γ production was selective rather than reflecting a general shut down in cytokine translation, as all the targeted cells showed a much smaller, albeit significant, reduction in TNF α production.” and “Taken together, these results indicate that lack of fully matured eIF5a does not impair TNF α mRNA transcription and has a very mild effect on its translation. By contrast, mature eIF5a is required both for transcription and translation of IFN γ .” No assays were used to support that translation of IFN γ or TNF α were affected.

We were careful to assess transcription of genes of interest by qPCR (Fig 2 & 6) and also more globally by mRNA-Seq in the GC7 treated cells (Fig 5) specifically because we wanted to be sure whether proteins were being impacted translationally by loss of eIF5a or its hypusination, rather than transcriptionally. We focussed our analysis on protein pathways that changed in protein abundance but had unchanged mRNA (from the GC7 RNA-Seq data, Fig 5a) and verified that mRNA was unchanged for proteins of interest in Eif5a, Dohh, and Dhps KO samples compared to controls by qPCR (Figs 3c, 6c and S4b). We are confident we are looking at changes in translation as our primary analysis was on newly-labelled nascent proteins in the mass spec analysis.

- It is important to assay the impact on autophagy in the experiments wherein the authors use spermidine, modulate eIF5A, and use GC7. As noted by the authors in the text, but not experimentally, spermidine can promote autophagy. Studies have shown that spermidine can promote autophagy via eIF5a, in other contexts (PMID: 31474573), which promotes the synthesis of the autophagy transcriptional regulator TFEB. How do the authors explain not seeing autophagy as an identified pathway in their list of translationally regulated genes affected by GC7 and eIF5aKO (Figure 4). Can a difference in autophagy status (autophagic flux) be detected when the eIF5a axis is genetically or pharmacologically modified, and thus help to explain the observed discrepancies in phenotypes of GC7-treated cells and the cells lacking eIF5a or DHPS?

This is an interesting point and it is entirely possible that we did not see autophagy as a changed pathway because we constrained our comparison of the nascent proteomics between KO and control cells to only those proteins which we could be sure were unchanged transcriptionally (using the GC7 RNAseq data). This meant we ended up with rather few (86) bona fide translationally regulated proteins which limits what gets picked out in pathway analysis. Within the nascent proteomes Tfeb was unchanged between KO and control cells (Table S1). In response to the reviewer, we have now assayed autophagy of our KO cells directly (new Fig 2c). Interestingly, equivalent decreases in autophagic flux were detected in the GC7 and eIF5a KO cells, so are unlikely to explain differences between these populations whereas we did not detect increased autophagic flux in Dhps or Dohh KO cells.

Reviewer #3 (Remarks to the Author):

The manuscript by Tan et al investigates the role of translation factor eIF5a in murine primary OT-I T cells. eIF5a is amongst the top 20 most abundant proteins expressed in murine cytotoxic CD8 T cells according to previous work (Hukelmann et al NI 2015) and there is evidence that protein translation plays a key role in the regulation of protein expression in activated and memory T cells (eg. Salerno et al NI 2018; Wolf et al NI 2020).

The authors used CRISPR-Cas9 in OT-I cells to target eIF5a as well as the enzymes (DHPS and DOHH) involved in its hypusination, a post-translational modification that is apparently necessary for its activity in facilitating peptide bond formation. The authors demonstrate substantial reductions in global protein synthesis, T cell proliferation and IFN γ production as well as a minor diminution in TNF α . The effects produced by targeting eIF5a were partially replicated by Dhps deletion and to a lesser extent in Dohh KO cells. LC-MS was then used to profile the T cell proteome, which identified several proteins involved in cell proliferation as well as transcription factors such as Irf4 and Tbx21 that are known to play roles in CD8 T cell activation and differentiation all of which were reduced in eIF5a KO T cells.

It appears that Infg mRNA is also reduced in eIF5a KO cells (Fig. 2C) and there may be similar reductions in Irf4 and Cdk1 mRNA (Fig. 5B), although the data for Irf4 and Cdk1 is highly variable and requires additional data points.

We provide newer qPCR data in Fig 6b and Fig S4b in which we have also included the eIF5a KO, as by combining 2 guides the KO efficiency (>80%) was similar to the Dhps and Dohh KOs so that we did not need to fix and sort cells. These data support our contention that the KOs do not reduce either Tbet or IRF4 mRNA production unlike Eomes and CDK1 transcription which is impacted.

It is also not clear why the Gene/Actb ratios for Irf4 and Cdk1 are very different between control samples (Thy1 KO and WT) in Fig 5B. There is a similar discrepancy in Fig. 2C for detection of TNF α RNA (compare Thy 1KO and WT ratio).

The Thy1 gene is a convenient control to monitor KO efficiency as it targets well and is a surface marker. Although cross linking Thy1 can activate T cells as it is GPI-anchored, it was not well documented that the KO had impacts in mature T cells, although Thy1 KO thymocytes, which express much higher levels of Thy1 than peripheral T cells, were shown to have augmented signalling (Hueber, A.-O., et al 1997. Curr. Biol. 7: 705). Having used this KO extensively we are now of the view it is not entirely neutral (as demonstrated by enhanced killing in Fig 3d). However we struggle to

think of another T cell expressed molecule that, when deleted, we could be certain would have no impact on T cell behaviour. Overall there were fewer changes in gene expression in the Thy1 KO compared to mock controls, so for many parameters it serves as an adequate control.

If eIF5a activity has an impact on transcription and/or stability of these transcripts, what are the mechanism responsible for this effect?

We think eIF5a acts indirectly in this regard through the altered translation of transcription factors, as demonstrated by loss of IRF4, Tbet and Eomes protein which are critical TFs for IFN γ production

It would also have been interesting to evaluate the effect of eIF5a on translation of repressed mRNAs eg. 5'TOP containing transcripts that are translated following mTORC1 signalling.

We found proteins encoded by 5'TOP mRNAs (genelist obtained from 10.1073/pnas.1912864117) overall were slightly yet significantly more produced in Eif5a KO cells than bootstrapped proteins, we have commented on this in the discussion (Line 401-403).

Finally, whilst the data clearly point to defects in cell proliferation and IFN γ expression, the findings do not put the role of eIF5a in context of other proteins of the translational machinery.

It is correct we did not focus on the consequences of loss of eIF5a on the other proteins of the translational machinery and agree this would be an interesting question, but feel it is somewhat outside the scope of the present study.

REVIEWERS' COMMENTS

Reviewer #1 (Remarks to the Author):

The authors answered all my points in a satisfactory manner.

Reviewer #2 (Remarks to the Author):

Dr. Zamoyska and team should be commended for their newly revised manuscript, which has largely addressed the points which I raised, and those of other reviewers. However, in the rebuttal they state "We are confident we are looking at changes in translation as our primary analysis was on newly-labelled nascent proteins in the mass spec analysis." I suggest that the authors consider toning down, or at least discussing the short comings of the Mass spec based approach used to draw conclusions related to mRNA translation. While I appreciate the approach to quantify nascent protein synthesis by Mass Spec, especially given the use of primary cells, to my knowledge this method does not provide the same depth as ribosome profiling (Ribo-seq). Thus the results obtained are related to translation, but this has not been formally tested in this paper.

Reviewer #3 (Remarks to the Author):

My previous concern that the findings do not put the role of eIF5a in the context of other proteins of the translational machinery remains unanswered.

If the authors have additional information comparing their findings with those from cells lacking the other elongation factors, I think that this information would reveal the contextual role of eIF5a.

Reviewer #1 (Remarks to the Author):

The authors answered all my points in a satisfactory manner.

Reviewer #2 (Remarks to the Author):

Dr. Zamoyska and team should be commended for their newly revised manuscript, which has largely addressed the points which I raised, and those of other reviewers. However, in the rebuttal they state "We are confident we are looking at changes in translation as our primary analysis was on newly-labelled nascent proteins in the mass spec analysis." I suggest that the authors consider toning down, or at least discussing the shortcomings of the Mass spec based approach used to draw conclusions related to mRNA translation. While I appreciate the approach to quantify nascent protein synthesis by Mass Spec, especially given the use of primary cells, to my knowledge this method does not provide the same depth as ribosome profiling (Ribo-seq). Thus the results obtained are related to translation, but this has not been formally tested in this paper.

We take the reviewer's point and have added the following sentence in the appropriate results section which will hopefully address their concern:

"The advantage of this approach is that it directly measures the synthesis of new proteins in order to characterise the functions of translation factors more sensitively than total proteomics, however, it may not be as comprehensive at measuring mRNA translation as conventional techniques such as ribosome profiling."

Reviewer #3 (Remarks to the Author):

My previous concern that the findings do not put the role of eIF5a in the context of other proteins of the translational machinery remains unanswered.

If the authors have additional information comparing their findings with those from cells lacking the other elongation factors, I think that this information would reveal the contextual role of eIF5a.

The reviewer makes an interesting point and we have compared our dataset with a published study in which eEF2 was knocked down in vivo and in vitro. eEF2 knockdown caused a general upregulation of ribosome associated proteins and there were some similarities between the nascent proteome analysis from *Eif5a* KO T cells and eEF2 knockdown cells, such as upregulation of the translation factor eEF1b and downregulation of eIF5, however, many more proteins were up- or down-regulated specifically in each dataset so there was not a great deal of overlap between them. For example, for the upregulated ribosomal proteins: Rps11, Rpl21, Rpl8, Rpl36, Rps25, Rps14, Rps4x, Rpl13, Rpl11, Rps3, Rps7, Rpl29, Rpl23, Rps19, Rpl12, Rps2, Rps10 were uniquely upregulated by reduction of eEF2 while Rps26, Rpl18, Rpl22, Rpsa, Rps5, Rps28, Rps15a, Rplp2, Rpl10a, Rpl24, Rpl19, Rpl27a, Rps20 were uniquely upregulated in the nascent eIF5a proteome and only Rpl5, Rpl30, Rps3a, Rpl31, Rpl17, Rps13, Rpl23a were common to both. Similarly there were few changes to translation factors in common between the two datasets. Furthermore

we are concerned that this comparison is not very robust as the different datasets were acquired differently, i.e. ours are newly synthesised proteins in a 2h window rather than total proteomics. Overall we believe that a dedicated study is required to really address this issue and this is outwith the scope of our current study, but we have commented on the eEF2 knockdown data in the discussion.